# A multi-proxy assessment of terrace formation in the lower Trinity River valley, Texas

Hima J. Hassenruck-Gudipati[1], Thaddeus Ellis[1], Timothy A. Goudge[1], David Mohrig[1]

[1]Department of Geosciences, Jackson School of Geosciences, The University of Texas at Austin, Austin, 78712, USA

*Correspondence to*: Hima J. Hassenruck-Gudipati (himahg@utexas.edu)

**Abstract.** A proposed null hypothesis for fluvial terrace formation is that internally generated or autogenic processes such as lateral migration and river-bend cutoff produce variabilities in channel incision that lead to the abandonment of floodplain segments as terraces. Alternatively, fluvial terraces have the potential to record past environmental changes from external forcings that include temporal changes in sea-level and hydroclimate. Terraces in the Trinity River valley have been previously characterized as Deweyville groups and interpreted to record episodic cut and fill during late Pleistocene sea-level variations. Our study uses high-resolution topography of a bare-earth digital elevation model derived from airborne lidar surveys along ~88 linear km of the modern river valley. We measure both differences in terrace elevations and widths of paleo-channels preserved on these terraces in order to have two independent constraints on terrace formation mechanisms. For 52 distinct terraces, we quantify whether terrace elevations fit distinct planes – expected for allogenic terrace formation tied to punctuated sea-level and/or hydroclimate change – by comparing variability in a grouped set of Deweyville terrace elevations against variability associated with randomly selected terrace sets. Results show Deweyville groups record an initial valley floor abandoning driven by allogenic forcing, which transitions into autogenic forcing for the formation of younger terraces. For these different terrace sets, the slope amongst different terraces stays constant. For 79 paleo-channel segments preserved on these terraces, we connected observed changes in paleo-channel widths to estimates for river paleo-hydrology over time. Our measurements suggest the discharge of the Trinity River increased systematically by a factor of ~2 during the late Pleistocene. Despite this evidence of increased discharge, the similar down-valley slopes between terrace sets indicate that there were likely no increases in sediment-to-water discharge ratios that could be linked to allogenic terrace formation. This is consistent with our elevation clustering analysis that suggests younger terraces are indistinguishable in their elevation variance from autogenic terrace formation mechanisms, even if the changing paleo-channel dimensions might, viewed in isolation, provide a mechanism for allogenic terrace formation. Methods introduced here combine river-reach scale observations of terrace sets and paleohydrology with local observations of terraces and paleo-channels to show how interpretations of allogenic versus autogenic terrace formation can be evaluated within a single river system.

## 1 Introduction

River valleys commonly contain fluvial terraces representing segments of older floodplain that are now located at elevations distinctly above the modern floodplain. These terraces sometimes preserve paleo-channels, or remnant river-channel segments. For exceptionally preserved features, channel widths, depths, bend amplitudes and wavelengths, and grain size record a signal of past river hydrology. Terrace formation requires net river incision that can be allogenically driven by tectonic uplift, sea-level fall, and/or modifications to water and sediment discharge via climate change or land-use change, including dam construction (Bull, 1990; Hancock and Anderson, 2002; Mackey et al., 2011; Pazzaglia, 2013; Womack and Schumm, 1977). What is more controversial is the character of the trigger that leads to the relatively discrete transfer of a section of active floodplain or valley floor into an inactive terrace or set of terraces elevated above flood height. In particular, can terraces formed by a punctuated sea-level fall, tectonic uplift, or sediment-to-water flux change be accurately separated from terraces formed by lateral migration and incision connected with the autogenic processes of river channel migration and channel-bend cutoff? Here we use attributes of terraces and their preserved paleo-channels in the coastal Trinity River valley in order to evaluate the likelihood of allogenic versus autogenic triggers driving terrace formation for previously established groups of Deweyville terraces (Bernard, 1950; Blum et al., 1995). Understanding how these terraces were most likely formed will help to constrain interpretations of the input signals for downstream deltaic deposits, which are recognized to embed both allogenic and autogenic signals (Guerit et al., 2020).

Commonly invoked allogenic triggers connected with terrace formation are (1) punctuated decreases in sediment-to-water flux that are assumed to embed a signal of regional climate change and (2) punctuated base-level fall controlled by either sea-level fall or tectonic uplift, all of which can drive periods of increased vertical incision along an extended length of river channel (Blum et al., 1995; Blum and Törnqvist, 2000; Bull, 1990; Daley and Cohen, 2018; Hancock and Anderson, 2002; Merritts et al., 1994; Pazzaglia, 2013; Pazzaglia and Gardner, 1993; Rodriguez et al., 2005; Wegmann and Pazzaglia, 2002). These focused periods of downcutting are interpreted to produce a spatially extensive terrace, or set of terraces, that preserve a fraction of the active fluvial surface and its river channel at the time of the terrace-forming event (Bull, 1990; Molnar et al., 1994; Pazzaglia, 2013; Pazzaglia et al., 1998). One expected morphology for terraces formed by allogenic triggers are extensive terraces flanking both sides of the river at a similar elevation, which would be expected during synchronous river incision. However, it is important to realize that the extent and pairing of these terraces can be substantially modified during ongoing valley incision and that unequal channel migration during relatively slow incision rates can produce similar characteristics (Limaye and Lamb, 2016; Malatesta et al., 2017).

Both theory (Parker et al., 1998a; Wickert and Schildgen, 2019) and experiments (Tofelde et al., 2019; Whipple et al., 1998) have shown how the long profile of a fluvial valley is set by the ratio of sediment-to-water discharge. Decreases in water-to-sediment flux led to slope increases via alluviation. Conversely, increases in water-to-sediment flux produce lower slopes through channel incision and valley formation. An allogenic trigger for terrace formation associated with paleohydrology change is therefore expected to produce a long profile for older terraces that are steeper than the long profile

of the younger and incising river. This reduction in slopes from older terrace sets to the modern floodplain has been observed
in both natural (Poisson and Avouas, 2004) and experimental (Tofelde et al., 2019) systems. Interestingly, a change in climate
that produced similar decreases or increases in both the water and sediment discharges would yield no change in the
downstream slope of the system and no episode of incision to drive terrace formation. Since water and sediment discharges
are strongly correlated within fluvial systems (Blom et al., 2017; Lane, 1955), it is quite possible that climate change might
not provide an allogenic trigger for terrace formation. If long profiles extracted from terrace sets are parallel to the slope of the
modern river than a different driver of incision must be at work. In the greater coastal zone this can be a base-level drop tied
to sea-level fall (Tofelde et al., 2019). For this reason it is tempting to use interpreted sets of subparallel terraces as a proxy
record for fluctuations in sea-level through time (Blum et al., 1995; Blum and Törnqvist, 2000; Merritts et al., 1994; Rodriguez
et al., 2005).

It has also been shown that terraces can form by autogenic processes that drive spatially variable incision rates under

conditions of persistent, allogenically forced base-level fall (Bull, 1990; Finnegan and Dietrich, 2011; Limaye and Lamb,
2014; Merritts et al., 1994; Muto and Steel, 2004; Strong and Paola, 2006). Autogenic terraces can be produced by channel
narrowing (Lewin and Macklin, 2003; Muto and Steel, 2004; Strong and Paola, 2006) and river-bend cut off (Erkens et al.,
2009), both of which can increase bed incision rates via upstream propagating knickpoints (Finnegan and Dietrich, 2011).
Processes that lead to terraces that have autogenic characteristics include local variations in channel dynamics, channel bed
slope, and sediment contribution from tributaries (Erkens et al., 2009; Lewin and Macklin, 2003; Womack and Schumm,
1977). In particular, river bend cut-off can locally increase the channel slope, driving channel-bed incision that transitions a
segment of floodplain into a terrace (Erkens et al., 2009; Finnegan and Dietrich, 2011). This autogenic trigger produces terrace
heights consistent with elevation drops associated with bend cutoffs (Finnegan and Dietrich, 2011). An additional autogenic
process that can trigger terrace formation is variable rates of lateral channel migration during persistent base-level fall (Lewin
and Macklin, 2003; Limaye and Lamb, 2016). Both unsteady lateral migration and bend cut-off preferentially generate terraces
that host only a small number of paleo-channel bends (Finnegan and Dietrich, 2011).

Here we present a study of three previously classified sets of fluvial terraces composing the Deweyville

allostratigraphic units of the lower Trinity River valley that have previously been described occurring at three distinct elevation
trends (Bernard, 1950; Blum et al., 1995; Heinrich et al., 2020; Young et al., 2012). These terraces have been interpreted as
forming in response to allogenic triggers that include Pleistocene sea-level fluctuations and climate-controlled changes in
water-to-sediment discharge (Anderson et al., 2016; Blum et al., 2013, 1995; Blum and Aslan, 2006; Rodriguez et al., 2005;
Saucier and Fleetwood, 1970). We analyze whether these purported allogenic triggers can be distinguished from a null
hypothesis that terraces were formed by autogenic processes during long-term valley incision associated with persistent sea-
level fall during the Last Glacial Period (from the end of the Eemian to the Last Glacial Maximum). To do this we implement
a multi-proxy approach that (1) compares variability in terrace elevations for each classified set against elevation variability
for randomly selected terraces, (2) evaluates temporal changes in paleo-hydrology as defined by segments of paleo-channels
preserved on terrace surfaces, and (3) relates paleo-slopes defined by the terrace sets to the long profile of the modern Trinity
River. Our analysis reveals that the upper set of terraces is most likely the product of an allogenic trigger, while the lowest set
of terraces is most likely the product of autogenic processes. The formational driver for the third, intermediate set of terraces
is equivocal. This result documents how the study of terraces can be employed to substantially refine paleo-environmental
interpretations that are generated using these preserved fragments of relict landscapes.

## 2 Geological Setting

The Trinity River has the largest drainage basin contained entirely within the state of Texas, with an area of over
46,000 km$^2$. It flows from northwest of Dallas, Texas, to Trinity Bay, where it empties into the Gulf of Mexico. Our study area
is an ~88 linear-km stretch of the lowermost Trinity River valley from just north of Romayor, Texas, to just north of Wallisville,
Texas (**Fig. 1**). Prone to flooding, the Trinity River has a median peak-annual discharge of 1679 m$^3$/s at Romayor, TX (USGS
08066500) and 1484 m$^3$/s at Liberty, TX (USGS 08067000) for the 2000 – 2020 hydrograph (U.S. Geological Survey, 2020a,
2020b).

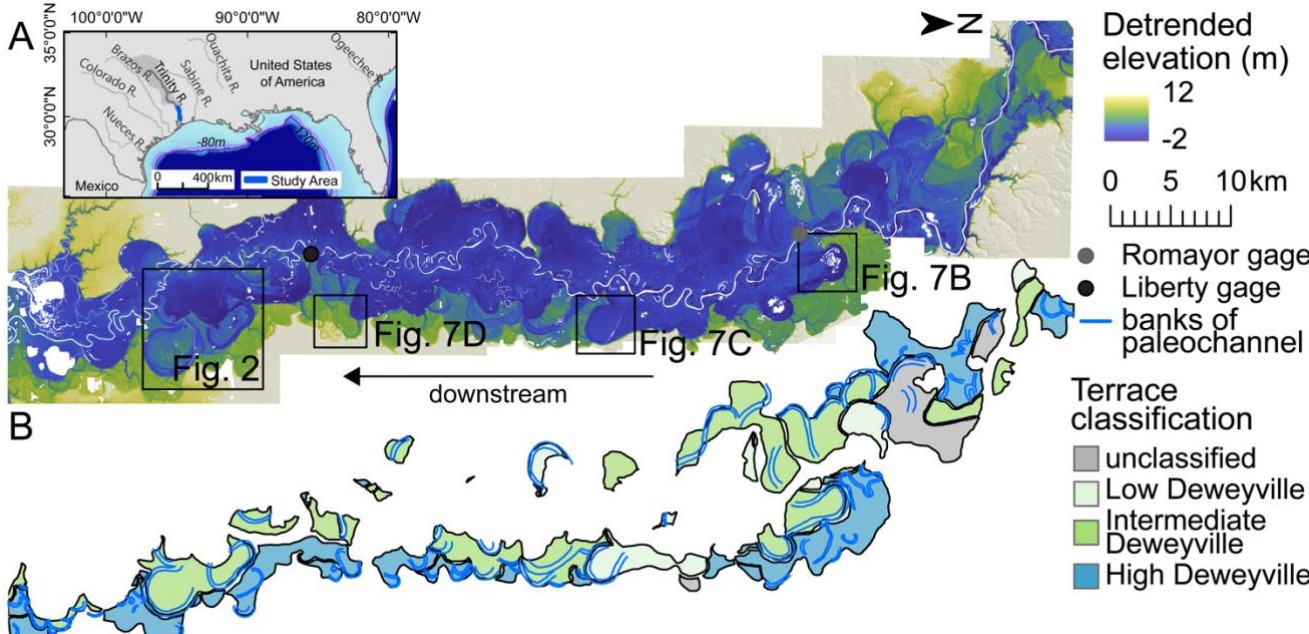


**Figure 1. (A) 2011 bare earth digital elevation model (DEM) from airborne lidar with (B) terrace and paleo-channel outlines of the Trinity River, Texas, valley. (A) The lidar DEM has been detrended using the modern valley slope to emphasize local elevation variability. The black boxes mark the extent of Fig. 2 and 7B-D. USGS gage stations at Romayor and Liberty are marked in grey and black, respectively. The downstream extent of the data is ~10 linear km upstream of the river outlet into the Trinity Bay of the Galveston Bay. (B) Terraces are preferentially distributed on the east of the valley.**

The Trinity River has been subject to climate and sea-level variations throughout the Quaternary (Anderson et al.,
2014; Galloway et al., 2000; Simms et al., 2007); however, the river catchment has never been glaciated and is interpreted to
have maintained an approximately constant drainage area over this time (Hidy et al., 2014). The lower Trinity River valley is
incised into the Beaumont and Lissie formations of Middle to Late Pleistocene age (Baker, 1995). Within the valley,
Deweyville terraces are post-Beaumont in age and formed prior to the Holocene (**Fig. 2A-B**). Age equivalent terraces with
preserved segments of large paleo-channels are also found in alluvial valleys ranging from Mexico to South Carolina and are
often classified as belonging to the same Deweyville bounding surface and allostratigraphic unit. Traditionally, the formation
of Deweyville terraces has been interpreted as the product of high frequency Pleistocene sea-level cycles (Anderson et al.,
2016; Bernard, 1950; Blum et al., 1995) with distinct episodes of incision and subsequent valley deposition (Blum et al., 2013;
Blum and Aslan, 2006). The history of climatic variation, lack of glaciation, and superb preservation of late Pleistocene terraces
make the lower Trinity River valley an ideal location to study terrace formation and to ask what processes these geomorphic
features record.

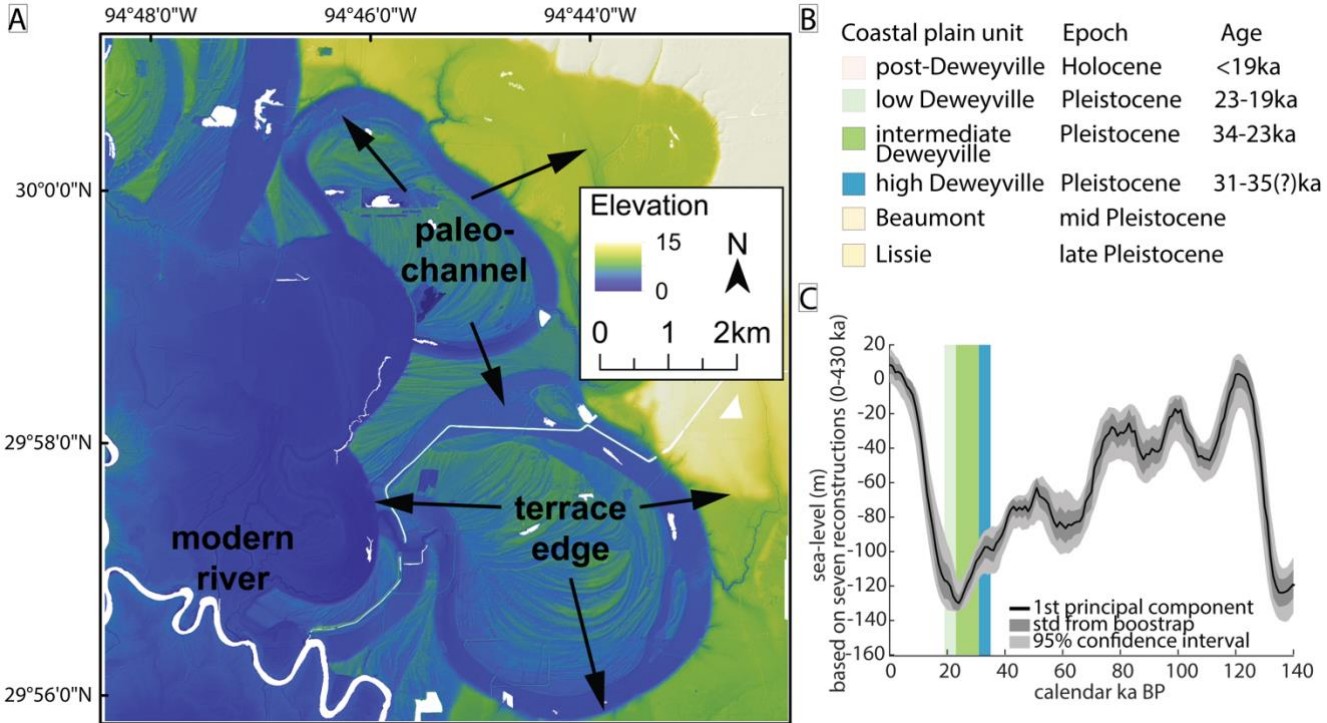

**Figure 2. (A) Morphological features of the Trinity River valley. Several terraces are preserved at different elevations with the black**
**arrows marking the edges of the terraces. The labelled paleo-channel has a width that is ~2 times the modern river channel width.**
**(B) Regional stratigraphic framework (Garvin, 2008; Blum et al., 2013). (C) Global sea-level from seven reconstructions based on**
(Spratt and Lisiecki, 2016) **and Deweyville teracce ages from (B) in light green, green, and blue.**

The Deweyville terraces are associated with bounding surfaces of three allostratigraphic units, and can be grouped

into three sets of terraces: high, intermediate, and low (Bernard, 1950; Blum et al., 1995; Young et al., 2012). Sea-level rise
during the Holocene has induced valley-floor sedimentation that has partially buried the low-terrace bounding surface (Blum
et al., 1995; Blum and Aslan, 2006). Age control for terraces in the lower Trinity River valley is limited to eight dates using
optically stimulated luminescence (OSL) (Garvin, 2008). Based on these data, Garvin (2008) reports an OSL age of 35 - 31 ka
for channel activity on high Deweyville terraces (N = 1), 34 - 23 ka for intermediate Deweyville terraces (N = 4), and 23 - 19
ka for low Deweyville terraces (N = 3). With only a single OSL date from the high Deweyville terraces, these features could
be as old as 60-65 ka based on existing stratigraphic frameworks (Blum et al., 2013).

The global sea-level curve shows an overall range of ~33m between 35 and 19ka (**Fig. 2C**, Spratt and Lisiecki, 2016).

The Pleistocene sea-level curve for the Gulf of Mexico during the period of Deweyville terrace formation shows high-
frequency variability superimposed on a longer-term net sea-level fall (Anderson et al., 2016; Simms et al., 2007). Between
35 and 19 ka, short-term rises and falls in sea-level are estimated to have been as large as 20 m and 60 m, respectively
(Anderson et al., 2016). Deweyville bounding surfaces have been interpreted to represent three discrete sets of terraces formed
during distinct oscillations in sea-level (Anderson et al., 2016; Bernard, 1950; Blum et al., 1995; Morton et al., 1996; Rodriguez
et al., 2005; Thomas and Anderson, 1994). The three sets of terraces also have been interpreted as recording episodes of relative
sea-level stasis with extensive lateral migration of the river channel, separated by punctuated incision tied to accelerated sea-
level fall (Blum et al., 2013; Blum and Aslan, 2006). The commonality between these two interpretations is an allogenic driver
for terrace formation.

Paleo-channels have long been recognized to record past hydrologic conditions and associated climatic variations

(Church, 2006; Knox, 1985). Terraces of the Trinity River valley preserve segments of abandoned river channels that range in
apparent widths and depths (**Fig. 2**). Previous researchers have interpreted increases in these paleo-channel widths and radii-
of-curvature for paleo-channel bends as products of increases in river discharge and precipitation (Church, 2006; Knox, 1985;
Saucier and Fleetwood, 1970; Sylvia and Galloway, 2006), and possible associated changes in vegetation and/or bank
erodibility (Alford and Holmes, 1985; Blum et al., 1995; Saucier, 1994). Paleo-channel morphologies thus provide a record of
external paleo-environmental change in the lower Trinity River valley that is independent of any signal encapsulated in terrace
formation. Therefore, using both terrace elevations and paleo-channels, we have two geomorphic proxies to compare and
contrast while assessing terrace formational processes among the Deweyville bounding surfaces.
**3 Null Hypothesis: Terrace Formation**

Following the proposal of Limaye and Lamb (2016), our null hypothesis for terrace formation is that punctuated

incision by autogenic triggers dominate terrace development. Only after formational mechanisms internal to the system have
been considered and rejected, should we consider allogenic triggers for terrace formation. Our method for testing the null
hypothesis acts to separate the regional expression of an allogenic driver from more localized terrace production by autogenic
processes. It is based on the observation that allogenic triggers produce synchronous, regionally extensive terraces that
approximately preserve surface elevations defining a single paleo-valley slope (Bull, 1990; Pazzaglia et al., 1998). It therefore
follows that a group of terraces formed by a contemporaneous allogenic trigger should preserve lower variability in elevations
about a best-fit plane estimating this paleo-slope than groupings of randomly selected terraces (Fig. XA). Conversely,
autogenically produced terraces preserve a multitude of elevations that we do not expect to define a contemporaneous long
profile (Fig. XB). Therefore, groupings of local autogenic terraces are expected to be indistinguishable from sets composed of
randomly selected terraces.

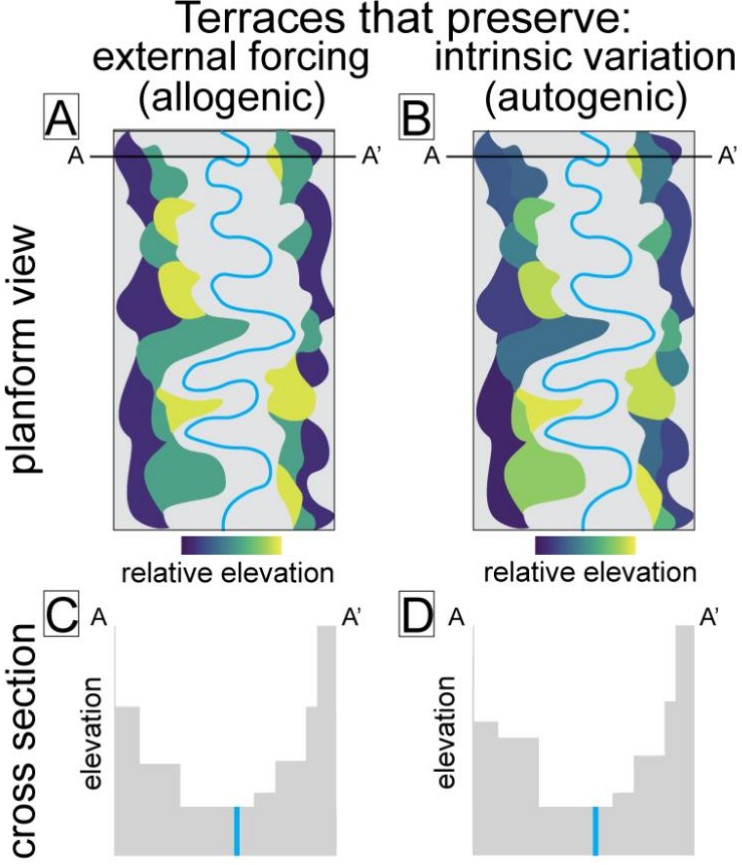


**Figure 3. Conceptual diagram showing distribution of terrace elevations expected in planform view (A-B) and cross sectional view**
**(C-D) for allogenic triggers (A, C) and autogenic triggers that form terraces (B, D). Here we show three distinct terrace sets for**
**autogenic triggers and an undistinguishable number of terrace sets.**
**4 Approaches and Observations**

Our study used elevation data derived from four airborne lidar surveys collected for the Federal Emergency

Management Agency (FEMA) and Texas' Strategic Mapping Program (StartMap) in 2011, 2017, and 2018 (FEMA, 2011;
StartMap, 2017a; StartMap, 2017b; StartMap, 2018). These four surveys were merged to produce a single bare earth digital
elevation model (DEM) with a 1 m grid spacing. The horizontal accuracies of the four original lidar point clouds from 2011,
2017a, 2017b, and 2018 are 0.6 m and 0.4 m, 0.25 m and 0.29 m, 0.20 m and 0.20 m, and 0.20 m and 0.20 m, respectively. All
data were referenced to the NAD83 horizontal datum.  The vertical accuracy for the original lidar point clouds from 2011,
2017a, 2017b, and 2018 are 0.4m, 0.29m, 0.20m, and 0.20m, respectively, and all data were refenced to the NAVD88 vertical
datum.
Individual terraces and paleo-channels were manually mapped on the merged DEM using ArcGIS. A terrace was
defined as a genetically similar surface that is offset in elevation from its surrounding topography. Previously, Blum et al.
(1995) mapped terraces on the Trinity River, which was extended by Garvin (2008), using a combination of satellite images
and DEMs. Based on these maps, terraces were classified as high, intermediate, or low Deweyville or marked as unclassified
if the surface had not been previously identified in Garvin (2008). Care was taken to only map the sections of terrace surfaces
that did not appear to be modified by later fluvial processes. Elevations defining each terrace were extracted from the DEM
using a 5 m grid resolution for a total of 164,520 measurements across all mapped terraces. A grid resolution lower than the
DEM resolution was selected to conserve available computational resources and to speed up analyses. Mapping on the 5-m
grid still produced hundreds of points for bare-earth elevation on each terrace, thereby producing estimates for the topography
that are comparable to calculations made using the full resolution DEM.
From these elevations, the median value and interquartile range were found for each of the 52 mapped terraces. Since
the Trinity River valley in the study area trends N-S, the elevation data for each terrace is plotted against median UTM northing
in **Fig. 4A**. The best-fit plane for the modern valley was used to generate detrended elevations for each terrace DEM
measurement by subtracting it from the spatially corresponding modern valley best-fit plane value. This plane defining the
modern valley floor was generated from a subsampled DEM with a 10 m grid resolution. The RMSE of the plane fit is 1.36 m
with most of the >4,500,000 points falling within 5m of the plane. Plotting the residuals to the best-fit plane along UTM
northing reveals some structure in the most downstream southern long profile extent (**Fig. 4A insert**), likely due to Holocene
sedimentation (Blum et al., 1995; Blum and Aslan, 2006). However, we do not think this affects our detrended terrace analysis.
The detrended median elevations and associated interquartile ranges for each terrace are presented in **Fig. 4B**. We then
compared the distributions of detrended elevations for the terrace classifications. Each classified distribution, scaled to its
contribution to the overall number of detrended elevations, is plotted in **Fig. 5**. Even though their median values are different,
the detrended elevation distribution for the intermediate Deweyville terraces fully overlaps with that of the low Deweyville
terraces (**Fig. 5**).

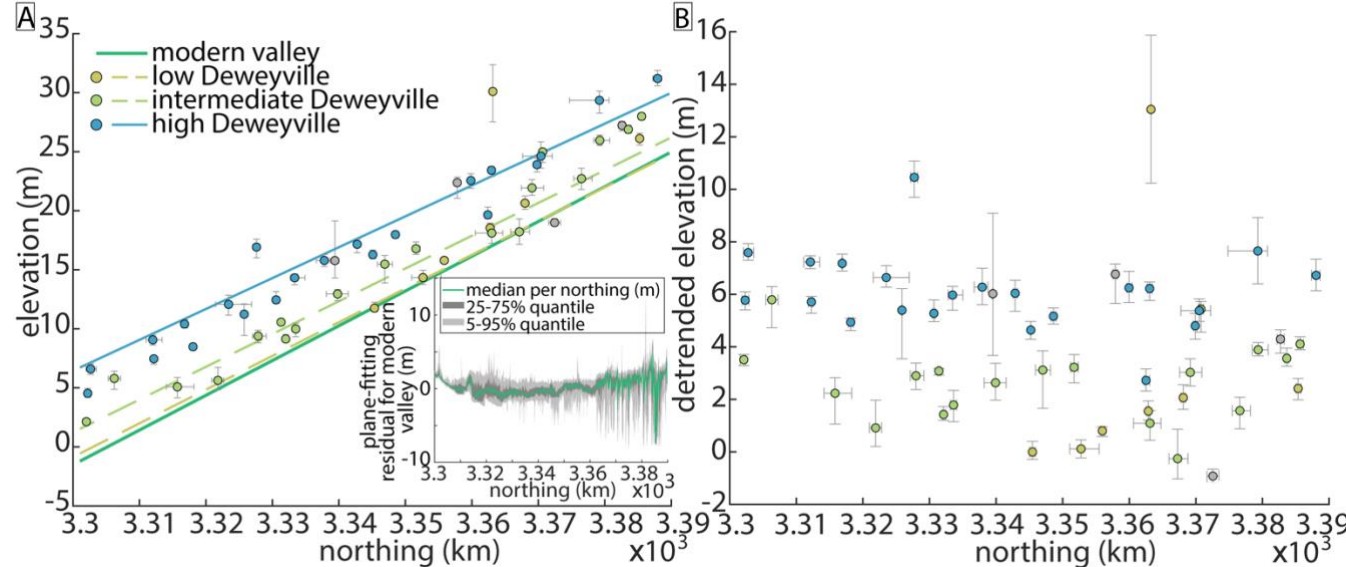


**Figure 4. (A) Median terrace elevation and (B) detrended elevation for the 52 terraces along the N-S trending valley, colored by**
**terrace category by Blum (1995) and Garvin (2008) as high, intermediate, and low Deweyville. Terraces not previously identify were**
**left grey. The error bars represent the interquartile range around the median terrace UTM and elevation values. The dark green**
**line corresponds to the plane fitted to the 10m DEM of modern valley elevations and the insert shows the residual of this plane fit.**
**Blue, green, and light green lines indicate the plane fit to 1m DEM terrace elevations assigned to each Deweyville terrace category.**

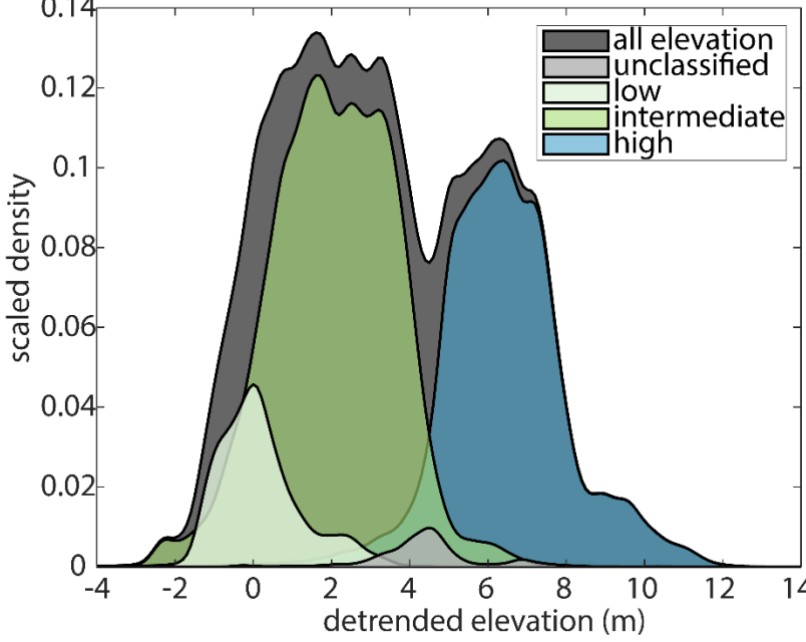


**Figure 5. Distributions of detrended elevations for terraces classified by Garvin (2008). Distributions were generated using a**
**Gaussian kernel with bandwidth = 0.2 and scaled by the proportion of the total elevation points (164,520) present in each**
**classification. There are 16,543, 84,784, 60,244, and 2960 points in the low, intermediate, high, and unclassified groupings,**
**respectively. There is complete overlap between the detrended elevations of low and intermediate terraces. Terraces classified as**
**high and intermediate have less overlap. The median detrended elevations for the low, intermediate, high, and unclassified**
**Deweyville groupings are 0.3m, 2.05 m, 6.3 m, and 4.41 m.**

## 4.1 Testing Terrace Formation using Elevation Data

We began our hypothesis testing by determining the best-fit plane to all of the raw elevation points (x, y, z) for terraces
classified into the three Deweyville groups by Blum et al. (1995) and Garvin (2008) using a linear least-squares method. A
planar surface was chosen for this analysis because the modern river-surface and valley profiles are near linear in our area of
study (**Fig. 4A**). The goodness of fit for these three planes to their associated terrace data was captured by the root-mean-
square error (RMSE), which provides a measure of average variability of actual terrace elevations about the best-fit plane (**Fig.**
**6**). The next step was to compare the properties of these fitted planes against planes fit to terraces randomly drawn from the
overall population. The randomly assigned terraces were put into one of three groups that had the same number of elements as
the classified high (n= 22), middle (n=19), and low (n=8) Deweyville terraces. Best-fit planes were calculated and their RMSE
was recorded. This process of randomly assigning terraces into three groups was then repeated 50,000 times in order to derive
a large dataset of elevation variability characterizing randomly grouped terraces (**Fig. 7**

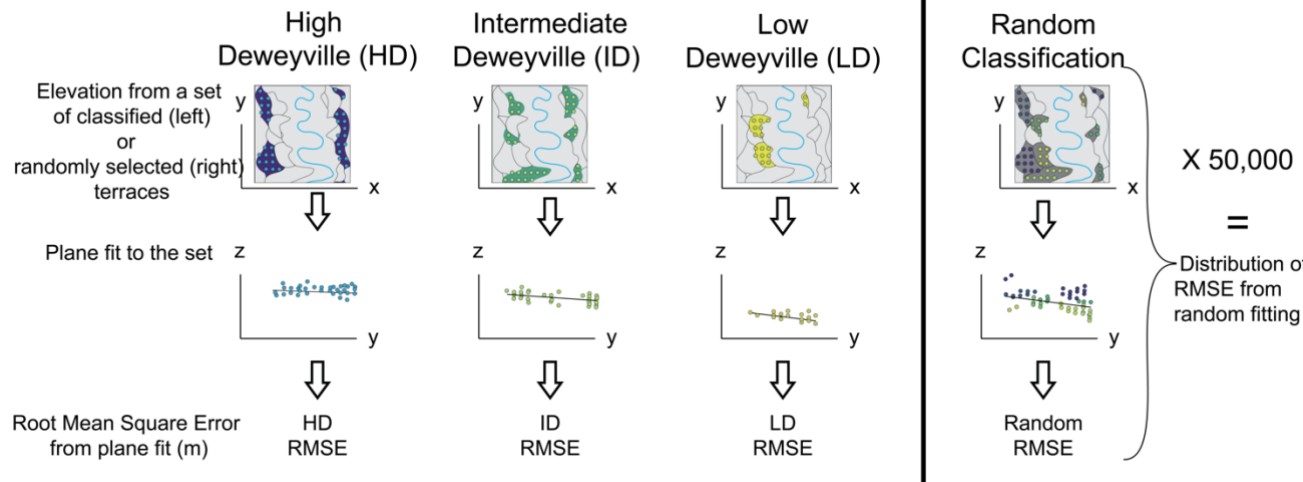

**Figure 6. Method to determine if classifications assigned to terraces represent distinct terrace groups. (Left) A plane was first fit to**
**elevations extracted from the classified terrace groups in Garvin (2008) at a 5 m grid resolution. (Right) We then fit planes to three**
**randomly grouped sets of terraces using the same elevation data, iterating 50,000 times, for a total of 150,000 fits. The root mean**
**square error (RMSE) of the plane fit from each of the previously classified terrace groups was compared to the distribution of RMSE**
**of the randomly grouped terraces (Fig. 7).**

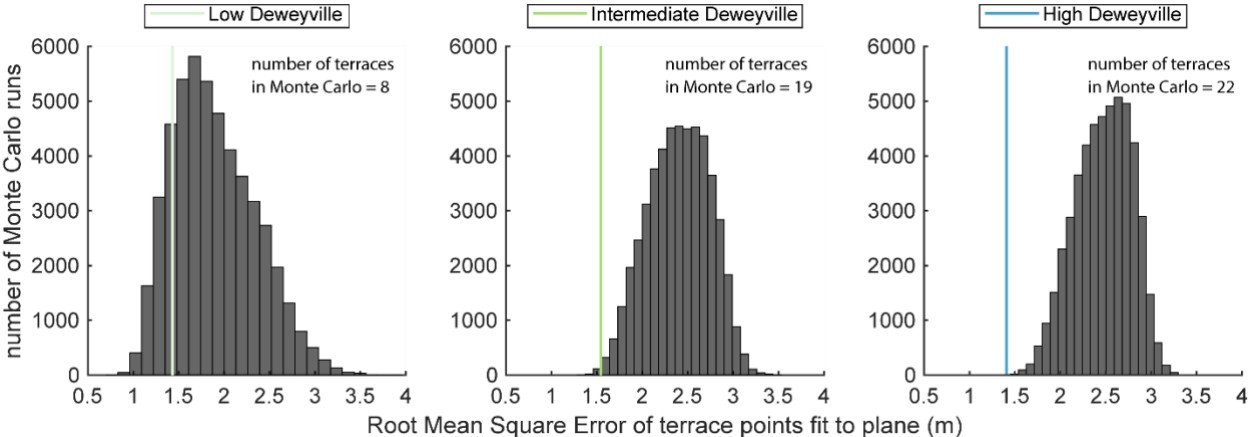


Figure 7. Root mean square error (RMSE) of a plane fit to elevation points of terraces previously classified as high Deweyville, intermediate Deweyville, and low Deweyville in the Trinity River valley compared to a distribution of RMSE from 150,000 randomly grouped terraces. The low, intermediate, and high Deweyville terrace sets have RMSEs of 1.43m, 1.54m, and 1.41m, respectively. All of the Deweyville classifications (light green, green, and blue lines) have an RMSE that falls within the distribution of RMSE for randomly grouped terraces. The high Deweyville classification is the closest to falling outside of the distribution, ~3.4 standard deviations away from the random terraces RMSE distribution mean of 2.47 m (22 terrace groupings). The low and intermediate Deweyville classification are ~1.0 and ~2.5 standard deviations away from the RMSE distribution mean of 1.89 m and 2.40 m for 8 and 19 terrace groupings, respectively.

**4.2 Evaluating Terrace Formation using Paleo-Channel Analysis**

Change in the discharge of the Trinity River during the late Pleistocene was estimated using the 79 mapped segments of paleo-channels preserved on terrace surfaces. Mean bankfull width ($B_{bf}$) for each paleo-channel mapped on the bare-earth DEM (**Fig. 1B**) was calculated from measurements extracted at 10 m intervals along each paleo-channel centerline (**Fig. 8**). Representative sidewall slopes (rise/run) for these paleo-channels range between 0.02 and 0.26 (**Fig. 8A**). These paleo-sidewall slopes fall within the range of modern sidewall slopes measured for the Trinity River in the study area by Smith and Mohrig, (2017, their Fig. 5). Therefore, we confidently use the paleo-channel widths extracted from the DEM without any correction to the widths associated with relaxation of the paleo-topography over time. These data were used to estimate a formative, bankfull discharge ($Q_{bf}$) for sand-bed rivers following the hydraulic geometry relationship developed by Wilkerson and Parker, (2011):

$$\frac{B_{bf}*g^{\frac{1}{5}}}{Q_{bf}^{\frac{2}{5}}} = 0.0398 * \left(\frac{D_{50}*\sqrt{R*g*D_{50}}}{v}\right)^{0.494\pm0.14} * \left(\frac{Q_{bf}}{D_{50}^2\sqrt{g*D_{50}}}\right)^{0.269\pm0.031}, \quad \text{(1)}$$

where $v$ is the kinematic viscosity of water, $R$ is the specific gravity of the sediment ($R = \frac{\rho_s - \rho}{\rho}$), $\rho_s$ is sediment density, $\rho$ is water density, $g$ is gravitational acceleration, and $D_{50}$ is the median grain size of transported bed material. We used a value of 2650 kg/m³ for $\rho_s$ and a range of paleo-channel grain sizes taken from Garvin (2008), who sampled both the lower and upper portions of bar deposits within preserved channel fills (**Table 1**). The uncertainty in estimated discharge was quantified for each paleo-channel using Monte Carlo simulation. For each run of the simulation, we sampled from: (1) normal distributions with the reported means and standard deviations for each exponent in **Eq. 1**; (2) a normal distribution for channel width using

its measured mean and standard deviation; and (3) a uniform distribution of grain sizes constrained by measurements from each classified terrace set (**Table 1**). This Monte Carlo simulation was run 50,000 times for each paleo-channel. Paleo-discharge estimates derived for the 79 channel segments preserved on terrace surfaces are plotted as a function of median detrended terrace elevation in **Fig. 9**.

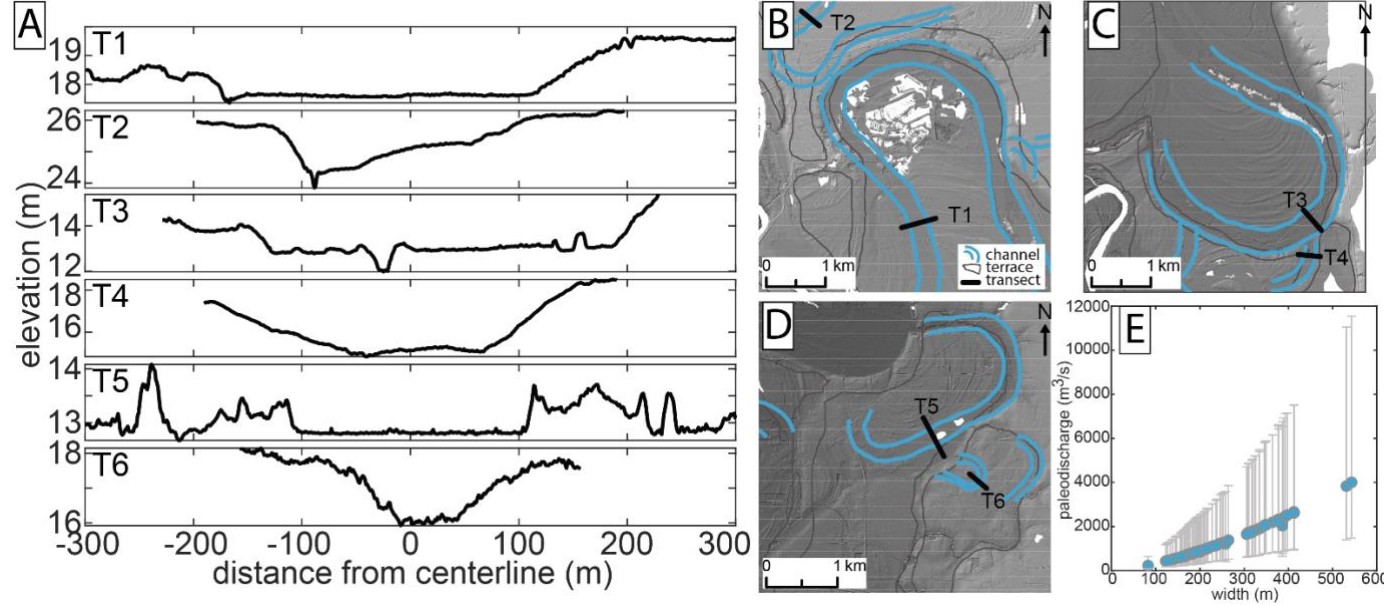

**Figure 8. Paleo-channel widths and paleo-discharge estimates. (A) Elevation transects for six paleo-channels (T1-T6). Transects are taken from locations indicated in (B)-(D) with mapped paleo-channels outlined in blue and terrace extents mapped outlined in grey. (E) Paleo-discharge estimates for the Trinity River are plotted as a function of their width. Each paleo-discharge was calculate using preserved channel width measurements and the discharge-width relationship from Wilkerson and Parker (2011) (Eq. 1). Error bars represent the first and third quartile of paleo-channel discharge estimates.**

| Garvin (2008) Terrace classification | Upper bar deposit lower range (mm) | Upper bar deposit upper range (mm) | Lower bar deposit lower range (mm) | Lower bar deposit upper range (mm) | Average grain size (mm) |
|---|---|---|---|---|---|
| Low Deweyville | 0.25 | 1.00 | 0.25 | 4.00 | 0.71 |
| Middle Deweyville | 0.125 | 1.00 | 0.50 | 2.00 | 0.59 |
| High Deweyville | 0.125 | 2.00 | 0.25 | 2.00 | 0.59 |

**Table 1: Grain size of terrace deposits from Garvin (2008), used for discharge calculations. The average grain size was calculated using the phi (logarithmic) scale.**

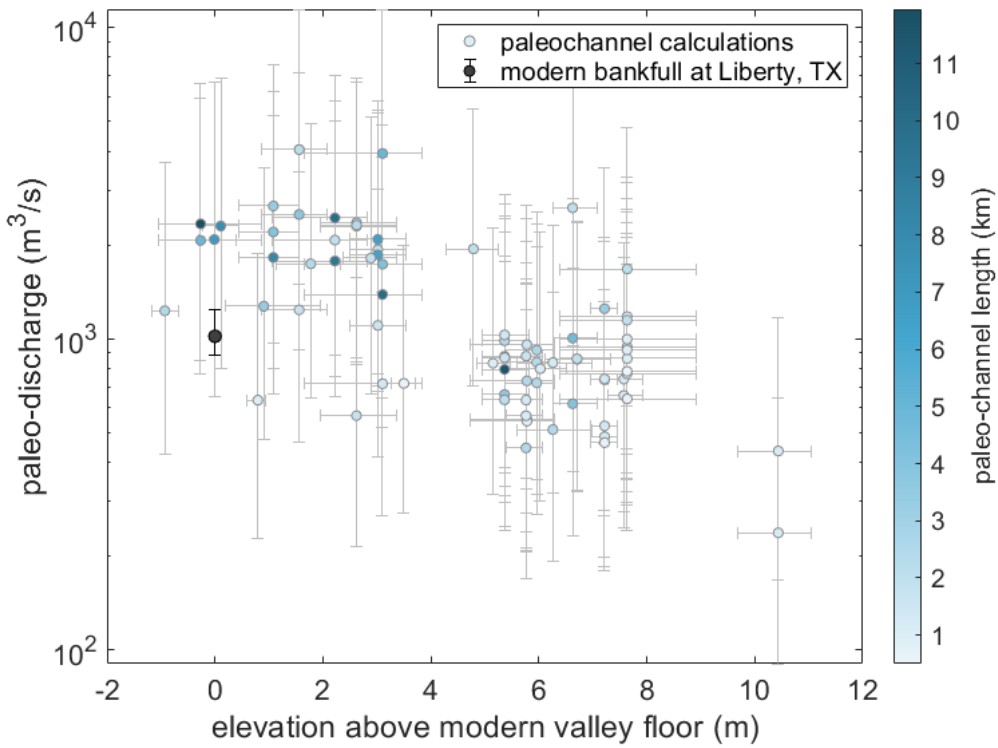

**Figure 9. Paleo-discharge estimates for the Trinity River plotted as a function of their associated detrended terrace elevations. Detrended elevations afford a crude stratigraphy for the discharges with the highest relative elevations representing older channels and lowest elevations representing younger channels. Each paleo-discharge was calculated using preserved channel width measurements and the discharge-width relationship from Wilkerson and Parker (2011) (Eq. 1). Error bars represent the first and third quartile of paleo-channel discharge estimates and terrace elevations above the modern valley. The symbol was shaded to the preserved length for each paleo-channel, with darker symbols equated to longer segments. The modern bankfull discharge at Liberty, TX was found using the methods described in the text, and plotted at 0m.**

Accuracy of the Wilkerson and Parker (2011) relationship for the Trinity River system was tested by calculating a $Q_{bf}$ value for the modern river channel and comparing it against the bankfull discharge logged at the USGS gage 08067000 at Liberty, Texas. The calculated bankfull discharge was estimated using the measured bankfull width of 170 m from the DEM at the gage site. The median particle size of bed material at Liberty has been measured at 200μm by the Trinity River Authority of Texas (Trinity River Authority of Texas, 2017). All other variables in **Eq. 1** were kept constant between the modern river and paleo-channels, yielding an estimate for the modern bankfull discharge of 830 m³/s. The reported residual standard error associated with the bankfull discharge **Eq. 1** (Wilkerson and Parker, 2011) was then used to approximate the error associated with this modern calculated bankfull discharge. The lower and upper standard error define a possible range between 340 and 2030 m³/s. These discharges estimated with **Eq. 1** compare favorably with the measured discharge found using the rating curve for the USGS Liberty gage station (https://waterdata.usgs.gov/nwisweb/get_ratings?file_type=exsa&site_no=08067000) and bank-line elevations for the swath of channel extending 300 m both upstream and downstream of the gage. The mean and

standard deviation of bank elevations for this swath was 7.68m and 0.34m, yielding a mean bankfull discharge of 1017 m$^3$/s
and discharges of 887 m$^3$/s and 1243 m$^3$/s corresponding to stages ±1 standard deviation in bank elevation.

## 297      4.3 Mixing Models and Bend Cutoff Analyses

To test the existence of statistical groupings within our terrace and paleo-channel data, a mixing model was used to generate
Gaussian mixture distributions that were fitted to both the 164,520 detrended terrace elevation points and the median discharges
estimated for the 79 paleo-channel segments (**Fig. 10**). Results are used to determine if Deweyville terraces should be divided
into three distinct sets. The Akaike Information Criterion (AIC) and Bayesian Information Criterion (BIC) were applied to
both mixing models in order to optimize the number of components used to represent each distribution (**Fig. 10C & 10D**).
Two and three components were selected for the distributions of detrended elevation points and median paleo-channel
discharges, respectively (**Fig. 10A & 10B**).

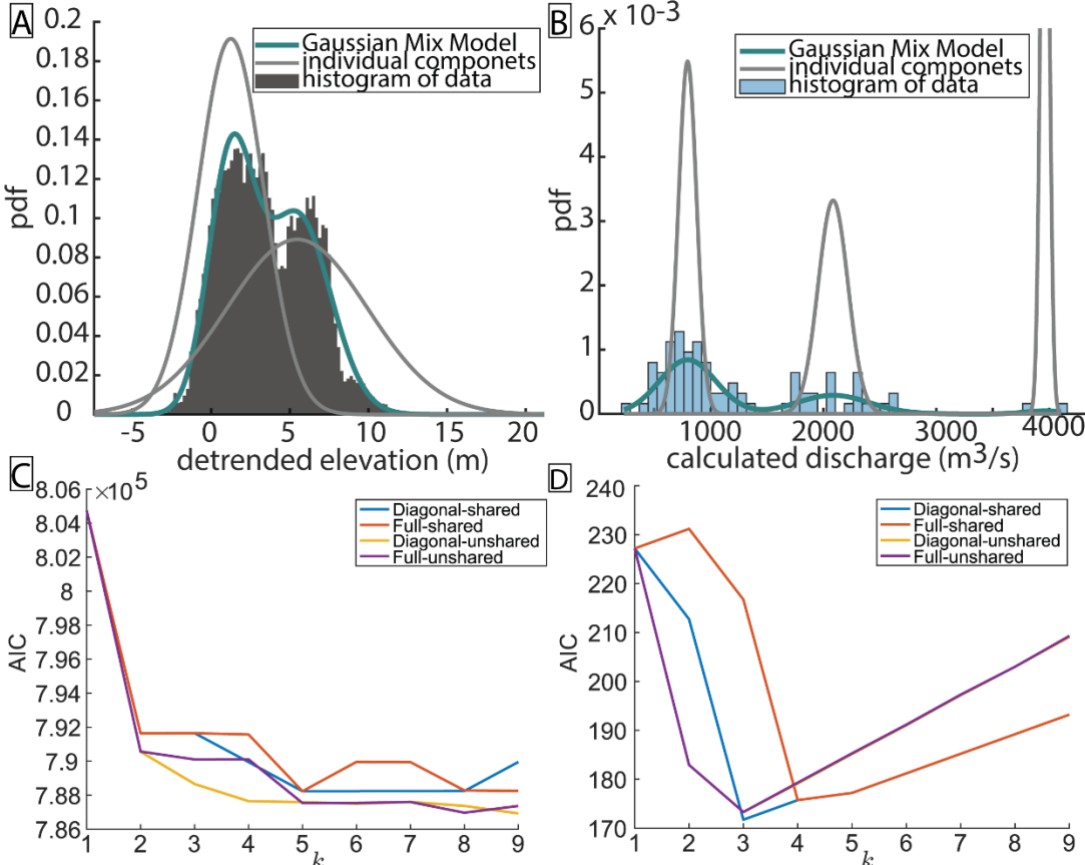


**Figure 10. Mixing model fits to measured distributions of terrace elevation and estimated paleo-discharges. Distributions using (A)**
**elevation and (B) paleo-channels support an interpretation of allogenic forcing for high terrace abandonment due to increasing**
**discharge. Akaike Information Criterion (AIC) for (C) detrended elevation and (D) paleo-discharge mixing model. BIC results are**
**not shown here but have similar trends to AIC. AIC results are shown for the mixing model that are solved for a diagonal and full**

 **covariance matrix and shared and unshared covariance. The model also used a small Regularization value to ensure the estimated**
**covariance matrix is positive.**
An important additional measurement used to assess whether terraces were abandoned due to enhanced local incision
driven by gradient change during channel bend cut-off was the elevation differences between 40 adjacent terraces. These
connections can first be assessed by comparing the minimum bounding box length of terraces, paleo-channel width, and paleo-
channel length (**Fig. 11**). These measured elevation differences between terraces were compared to estimated elevation changes
produced by bend cut-offs. We used **Eq. 2** to calculate the elevation drop produced by a bend cutoff as:

$$\Delta elevation_{bend\ cutoff} = length_{bend} * slope_{channel} \hspace{4cm} \textbf{(2)}$$

On several low, intermediate, and high Deweyville terraces, the lengths of paleo-channels that had one bend preserved were
measured using the bare-earth DEM (e.g., **Fig. 2**). The mean and standard deviations for bend lengths on the low, intermediate
and high terraces are $5.7 \pm 2.8$ km (n = 3), $4.6 \pm 3.0$ km (n = 10), and $2.3 \pm 1.1$ km (n =11), respectively. The overall distribution
above the modern valley floor of paleochannel lengths plotted in **Fig. 11B**. We approximated channel slope using the planes
fit to the terrace elevation points for each classification. The calculated mean slope and standard error for the low, intermediate,
and high terraces are $3.0 \times 10^{-4}$ ($3.1 \times 10^{-6}$), $2.9 \times 10^{-4}$ ($1.10 \times 10^{-6}$), and $3.0 \times 10^{-4}$ ($1.2 \times 10^{-6}$), respectively. Using Equation 2,
estimated elevation drops driven by a possible bend cut-off are 1.6 ± 0.8 m, 1.3 ± 0.9 m, and 0.7 ± 0.3 m (**Fig. 12A**).

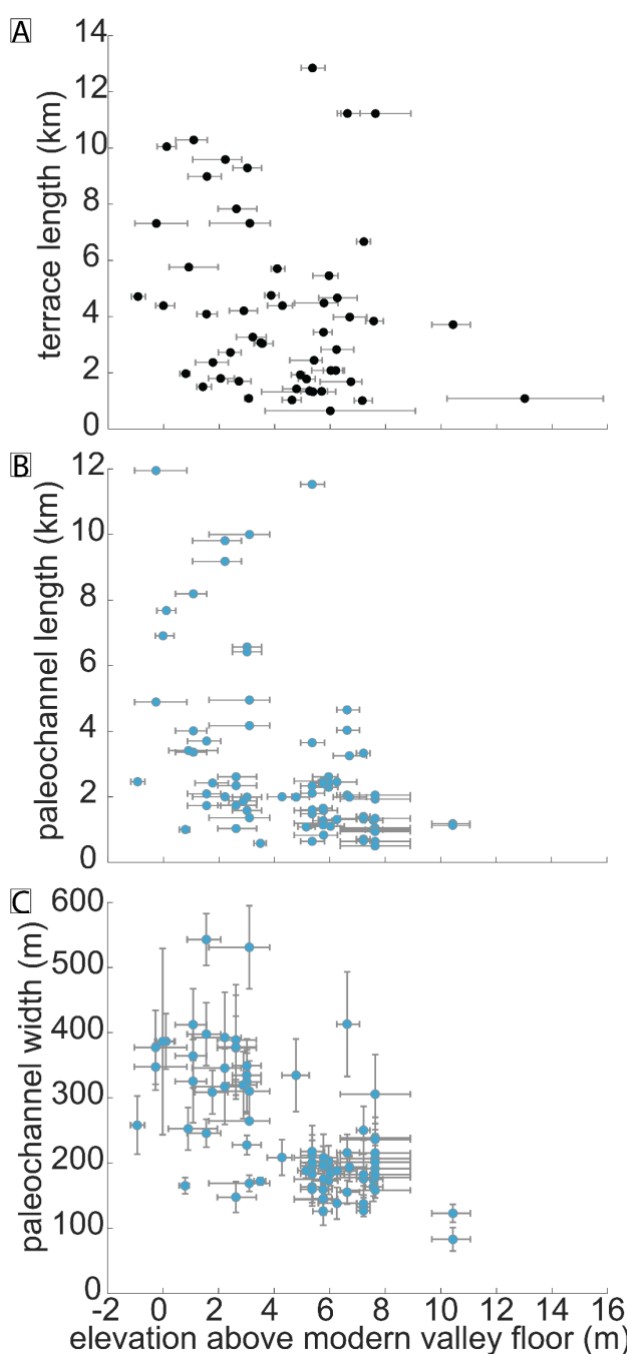


**Figure 11. (A) terrace length, (B) paleo-channel length, and (C) paleo-channel width plotted along the median elevation of the**
**associated terrace above the modern valley floor plane. The youngest terraces are more likely to have larger terrace lengths as well**
as paleochannel lengths. Terrace length was measured as the longest length of a minimum bounding box envelop for each terrace.
These envelopes were defined with edges in the N-S and E-W direction. Error bars show the interquartile range of each terrace.

329    An additional measurement used to evaluate the likelihood of terraces being produced by bend cut-off was the largest

number of channel bends present in a segment of paleo-channel preserved on a terrace surface. The number of channel bends
preserved on terrace surfaces can be used as an indicator for autogenic versus allogenic processes, whereby allogenic terrace
formation likely abandons larger paleo-floodplain sections, preserving multiple channel bends. For incising rivers, the
autogenic cut off of a single meander bend has been shown to be sufficient to produce the enhanced channel erosion required
to elevate relatively small sections of the previous active floodplain above flood levels (Finnegan and Dietrich, 2011).

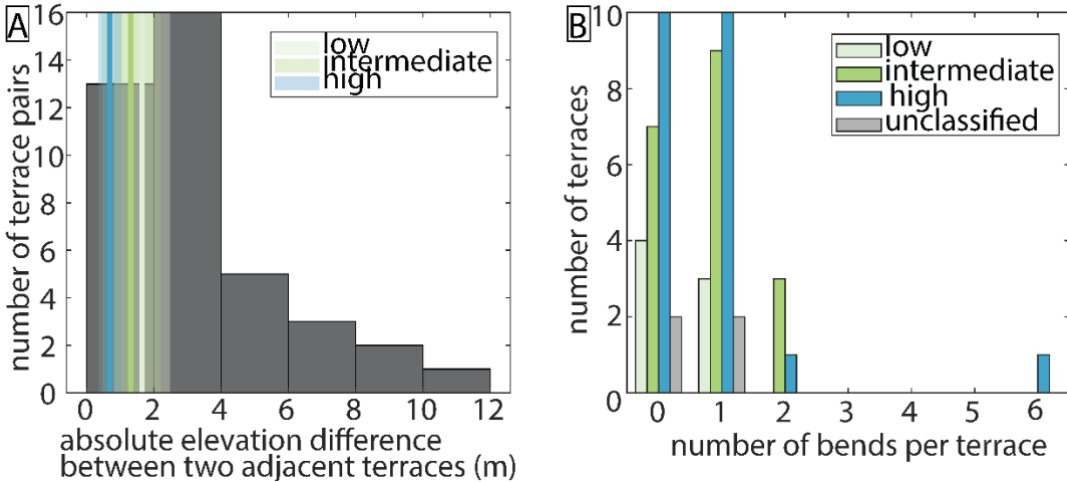


**Figure 12. Terrace properties used to assess the likelihood of meander bend-cutoff being the driver of terrace formation. (A)**
**Differences in elevation between adjacent terrace surfaces. Also plotted as vertical lines and swaths are the mean values ± 1 standard**
**deviation for elevation decreases expected from cutting off a single meander bend for paleo-channels of the low, intermediate, and**
**high Deweyville bounding surfaces. (B) Maximum number of paleo-meander bends preserved in a channel segment on each terrace.**
**Most terraces have between 0-1 channel bends preserved for one generation of channel. Only intermediate and high Deweyville**
**terraces have more than two channel bends preserved by a paleo-channel.**
**4. Summary of observations**

343    We mapped 52 terraces and 79 paleo-channel segments in the study area (**Fig. 1B**). Of these terraces, 22 are classified

as high Deweyville, 19 as intermediate Deweyville, 8 as low Deweyville, and 4 were left unclassified as they could not be
correlated with terraces mapped by either Blum et al. (1995) or Garvin (2008). The low, intermediate, and high Deweyville
terraces have median values for detrended elevations of 0.03 m, 2.06 m, and 6.37 m. Based on our mixture modeling, the mean
and standard deviation of the detrended elevation components are 5.6 ± 4.18 m and 1.32 ± 2.19 m with mixing proportions of
0.51 and 0.49, respectively. Similarly, the mean and standard deviation for the three Gaussian distributions describing paleo-
discharges are 795 ± 80 m$^3$/s, 2083 ± 139 m$^3$/s, and 4013 ± 21 m$^3$/s with mixing proportions of 0.68, 0.30, and 0.02,
respectively. Terraces vary in both size and shape, although they are typically elongate parallel to the valley axis and continuous
for less than 10 km in that direction. The distribution of terraces is asymmetric, with more terraces observed on the east side
of the valley (**Fig. 1B**). Consequently, most terraces are unpaired, meaning they have no topographic equivalent on the opposite
side of the valley.

The best-fit planes to elevations for the Deweyville terrace groups defined by Blum et al. (1995), Garvin (2008), and

Hidy et al. (2014) show remarkably similar slopes amongst terrace sets. The slopes and standard error for the low, intermediate,
and high terraces are $3.0\times10^{-4}$ ($3.1\times10^{-6}$), $2.9\times10^{-4}$ ($1.10\times10^{-6}$), and $3.0\times10^{-4}$ ($1.2\times10^{-6}$), respectively. These paleo-slopes are
indistinguishable from the estimated slope for the modern valley of $3.0\times10^{-4}$ ($8.0\times10^{-8}$) (**Fig. 4A**). It is not surprising that all
four profiles are well fit by planes given the fact that the studied river segment represents less than ten percent of the modern
river length and both grain size and discharge vary little over the studied reach.

Plane fits for the low, intermediate, and high Deweyville terrace sets have RMSEs of 1.43m, 1.54m, and 1.41m,

respectively. Values of RMSE for best-fit planes to randomly grouped terraces are sensitive to the number of terraces defining
a group. The fewer the number of terraces, the more likely it is that a low RMSE will result (**Fig. 7**). It is therefore important
when comparing randomly grouped terraces to the previously classified groups that the number of terraces in each be the same.
Running our analysis of 50,000 sets of randomly assembled terraces with the same number of elements as the low (n=8),
intermediate (n=19) and high (n=22) Deweyville groups yielded the following RMSE results. The median and interquartile
values of RMSE for planes fit to randomly selected terraces of the number present in the low-terrace classification are 1.82m,
1.55m, and 2.21m. The median and interquartile values of RMSE for planes fit to randomly selected terraces of the number
present in the intermediate-terrace classification are 2.41m, 2.15m, and 2.66m. And finally, the median and interquartile values
of RMSE for planes fit to randomly selected terraces of the number present in the high-terrace classification are 2.49m, 2.25m,
and 2.71m.

The RMSE values for the best-fit planes to the classified Deweyville terraces are plotted on their associated synthetic

RMSE distributions for randomly selected terraces in **Fig. 7**. Inspection of **Fig. 7** reveals little overlap between the classified
high terraces and the random samplings of terraces. For the high Deweyville case, there was only a 0.008% occurrence of
randomly selected terraces yielding an RMSE as low as 1.41m. A very different result was found for the classified low terraces,
where its RMSE falls well within the associated distribution of synthetic RMSEs with fully 21% of all randomly selected cases
having lower RMSE values. Minimal overlap was found between the RMSE for the classified intermediate terraces and the
distribution of RMSE values generated from random terrace groupings. Only 1% of the randomly selected sets terraces were
better fit to a plane than the classified group of intermediate terraces. Mapped paleo-channels have widths that range from 82
to 543 m (**Fig. 8**). Estimated bankfull discharges calculated using these widths (**Eq. 1**) range from 233 $m^3$/s to more than 4000
$m^3$/s (**Fig. 9**). These paleo-discharges cluster into two groups, one at lower discharges centered around $795 \pm 80$ $m^3$/s and one
at higher discharges centered on $2083 \pm 139$ $m^3$/s (**Fig. 10B**). The grouping of lower-discharge paleo-channels sit on terraces
that have median elevations >4.5 m above the modern valley floor and correspond to high Deweyville terraces (**Fig. 9**). The
grouping of higher-discharge paleo-channels is preserved on terraces that have median elevations from 0.2 m below to 5.2 m
above the modern floodplain and correspond to both intermediate and low Deweyville terraces (**Fig. 9**). The investigation of
paleo-channel characteristics revealed that paleo-channel widths, paleo-channel lengths, and overall terrace lengths all are
more likely to be greater for younger terraces (**Fig. 11**).  Most terraces have one or fewer channel bends preserved (**Fig. 12B**)
and only the intermediate and high Deweyville classifications possess terraces with more than two preserved channel bends.

## 5 Discussion and Conclusions

Late Pleistocene terraces of the lower Trinity River valley formed during a period of net sea-level fall punctuated by
shorter and smaller magnitude fluctuations (Anderson et al., 2016). Previous researchers have interpreted the formation of the
Trinity terraces, as well as those observed in other Texas coastal valleys, in the context of these fluctuations (Blum et al., 1995;
Blum and Aslan, 2006; Morton et al., 1996; Rodriguez et al., 2005). However, it has also been suggested that this terrace
formation in the lower Trinity River valley was driven by autogenic triggers (Guerit et al., 2020). The motivation for this study
was to develop tools to help distinguish between these two forcings that can produce terraces.
Several morphological characteristics exist to describe both the Trinity River terraces and their associated paleo-
channels. The terraces are most commonly unpaired (**Fig. 1**), which is expected during autogenic terrace formation associated
with unsteady lateral migration rates during formation (Bull, 1990; Merritts et al., 1994) and river bend cut-off (Finnegan and
Dietrich, 2011). On the flip side, paired terraces can also be formed during constant, albeit low vertical incision rates, during
lateral migration (Limaye and Lamb, 2016). Unpaired terraces can also be produced by unequal lateral river erosion post
terrace formation that preferentially removes half of a previously formed pair of allogenic terraces (Malatesta et al., 2017).
Similarly, lateral migration also affects the age distribution of the terraces preserved because younger terraces, closer to the
modern river are more likely to eroded away than older terraces (Lewin and Macklin, 2003; Limaye and Lamb, 2016). The
presence of unpaired terraces in the lower Trinity River valley may therefore be most indicative of the relative importance of
lateral migration for this system.
For the Trinity River, many of the valley-ward edges of the lower and intermediate Deweyville bounding surfaces
have the shapes of meander bends, recording the most outward extent of the active channel before the floodplain surface was
abandoned (**Fig. 1, Fig. 2**). We take this as evidence for the autogenic process of channel cutoff triggering terrace formation.
The observed elevation differences between adjacent terraces are also consistent with those predicted by cut off of a single
meander bend (**Fig. 12A**). Similar interpretations have also been made for strath terraces in bedrock (Finnegan and Dietrich,
2011). Furthermore, their tendency to be preserved as unpaired terraces with a small number ($< 2$) of channel bends is more
consistent with the stochastic nature of meander cutoffs by autogenic processes than large-scale incisional events due to
allogenic forcings (**Fig. 12A & 12B**, Finnegan and Dietrich, 2011). Therefore, the morphology of the Trinity River valley
terraces alone is suggestive of an autogenic forcing, but likely not sufficient to distinguish between allogenic versus autogenic
terrace formation.
We argue that a robust test for assessing the likelihood of autogenic versus allogenic forcing in terrace formation
comes from an analysis of the topographic variability of terrace sets inferred to have formed synchronously. Here we have
developed a method to quantitatively compare elevation variability of any classified group of terraces against randomly
selected terrace sets (**Fig. 6, Fig. 7**) so that we can evaluate whether a classified group is better organized than arbitrarily

selected ones. For the lower Trinity River valley, if the Deweyville terraces formed synchronously (Blum et al., 1995; Blum and Aslan, 2006; Morton et al., 1996; Rodriguez et al., 2005), one would predict that terraces within these groups would show lower variation about a best-fit plane than randomly grouped terraces (**Fig. 7**). Limaye and Lamb (2016) defined a unique elevation set as surfaces that are separated by more than 1m. They found that lateral migration during a constant incision rate versus pulsed incision rates can result in similar and indistinguishable terrace sets (Limaye and Lamb, 2016). Our approach builds on this idea and develops a framework that evaluates the magnitudes of variations in elevation amongst terraces compared to a fitted plane for the set. This approach is especially useful for studies where age control across terraces is not well constrained. Since we are assessing many elevation points from each terrace in the terrace set, it is possible to tease apart long profile variations for terrace sets only vertically separated by ~1m.

Our RMSE results show that the best-fit plane for the low Deweyville bounding surface cannot be separated from, and is instead consistent with, sets of randomly grouped terraces mimicking autogenic processes of either bend cutoff or of unsteady river lateral migration during constant base level fall (**Fig. 7B**). The driver for the intermediate Deweyville bounding surface cannot be unambiguously determined based on the RMSE analysis. The classified group is better organized than most, but not all, randomly generated groupings of terraces (**Fig. 7C**). The overlap leads us to presume that the null hypothesis of autogenic terrace formation cannot be robustly falsified. A different conclusion was reached for the high Deweyville bounding surface. With our RMSE analysis, we reject the null hypothesis of autogenic terrace formation. The high Deweyville bounding surface is most likely the product of punctuated allogenic change with an RMSE that is as small as any of the 50,000 values generated for random groupings of terraces (**Fig. 7A**). A difference between the low/intermediate versus high terraces was also found in the distribution of detrended terrace elevations using a 2 component Gaussian mixing model. The first component of this model overlaps with elevations classified as low and intermediate Deweyville, while the second component corresponds most closely to high Deweyville elevations (**Fig. 5, Fig. 10A**). We, therefore, conclude that the high Deweyville terraces are different than the other two sets and record an allogenic signal connected with early valley incision. This new analysis likely means that across a relatively short interval of time, <10 kyr, terraces on the Trinity River switched from recording an allogenic trigger in the high Deweyville bounding surface to being indistinguishable from terraces formed by autogenic triggers such as bend cut-off or unsteady lateral migration rates.

The connections between potential discharge changes and terrace formation were assessed using paleo-channel widths and grain size (**Fig. 9, Fig. 10B**). Paleo-channel discharge estimates reveal a factor of two increase in bankfull discharge moving from older, high Deweyville terraces to younger, intermediate, and low Deweyville terraces. The estimated changes through time in bankfull discharge are not matched by estimated changes in river long-profile or paleo-slope. Previously discussed best-fit planes to the Deweyville bounding surface have slopes that are roughly constant and indistinguishable from the modern long profile for the Trinity River (**Fig. 4A**). Theory by Parker et al. (1998) and experiments by Whipple et al., (1998) have demonstrated long-profile slope for sandy fluvial systems is a function of sediment-to-water discharges. Terraces associated with base-level fall have been shown to maintain consistent valley slopes (Tofelde et al., 2019). Experiments by the same authors also showed that sediment and/or water discharge changes produce changing slopes for terrace sets, which we

do not observe here. We suspect that the switch in discharge is not directly recorded in the terrace elevation because the change
in water discharge appears to have been approximately matched by a sediment-flux increase, as recorded in the constant long-
profile slope for the paleo-river. With no slope reduction, no incision would have occurred. As a result, discharge changes
recorded by segments of paleo-channels on the intermediate and low Deweyville terraces are not interpreted to have driven
incision and terrace formation. Instead, it likely that an autogenic trigger associated with persistent base-level fall drove the
terracing. Recent synthesis studies by Phillips and Jerolmack (2016) and Dunne and Jerolmack (2018) confirm that these
estimates of bankfull discharge are tied to moderate flooding and representative of mean climate properties. While our
estimated discharge changes over the latest Pleistocene are large, it is only half of the proposed four times increase reported
for similar paleo-channels preserved on terraces of the nearby lower Brazos River valley (Sylvia and Galloway, 2006).
Maintenance of a roughly constant slope while water discharge changed therefore almost certainly required
commensurate changes to sediment discharge. We can test this change in sediment discharge by looking at results from existing
studies. An increase in sediment discharge is in agreement with Anderson (2005), who suggests that sediment discharge was
greater during the LGM than today. However, calculations for the Trinity River by other authors do not currently reflect these
changes. Sediment discharges have been estimated to decrease during the LGM (intermediate and low Deweyville) based on
the BAQRT model by Syvitski and Milliman (2007) (Blum and Hattier-Womack, 2009; Garvin, 2008). Hidy et al. (2014) also
calculated [10]Be denudation rates and suggested that upstream weathering was greater during the interglacial periods and that
reworking of stored sediments was greater during glacial periods. However, Hidy et al. (2014) was not able to combine the
effects of reworking and upstream sediment flux using [10]Be to estimate the sediment discharge associated with terrace
formation. More recent methods were developed to estimate sediment discharge based on bedforms and stratigraphy, which
are exposed along the Trinity River (Mahon and McElroy, 2018). Therefore, there is also an opportunity to refine and improve
sediment discharge estimates for Deweyville terraces. Regardless, responsive adjustments to sediment discharge suggest that
throughout the latest Pleistocene, the river itself remained a predominantly transport limited system (Howard, 1980; Whipple,

475    2002).

Understanding the cut off of a river bend is important to identify autogenic triggers for terrace formation. We have
shown that a majority of Deweyville terraces in the Trinity valley preserve no more than a single paleo-channel bend (**Fig.**
**12B**) and that elevation differences between adjacent terraces are similar to an expected elevation change driven by channel
shortening through cut off to a river bend (**Fig. 12A**). These terrace properties highlight an opportunity for our community to
measure the number of bends involved in the autogenic shortening of river channels. Specifically, there is an opportunity to
quantify what percentage of cutoffs result in two or more bends being detached from the active channel in short amounts of
time, thereby refining an expected upper limit to the number of channel bends preserved on autogenically generated terraces.
Exceptionally preserved paleo-channels such as on the Trinity River, provide this opportunity to distinguish autogenic
processes responsible for terrace formation, and as such might provide a more faithful record of changes in discharge to the
system than terrace elevations and morphologies. An additional mechanism for reducing uncertainty in the processes that cut
Trinity terraces would be assembling a greater number of terrace ages. Increasing age control could constrain vertical versus
lateral migration rates for the river to a point where autogenic versus allogenic processes connected to terrace formation are
separable (Limaye and Lamb, 2016; Merritts et al., 1994).
Irrespective of terraces formation, other river systems across the southeastern United States have the potential to also
record a step-increase in formative discharge seen in the Trinity valley between the high to intermediate/low Deweyville
terraces. This change was likely driven by a wetter climate in southeast Texas during the period ~34–20 ka, based on OSL
dates for the low and intermediate Deweyville terraces (Garvin, 2008). During the Last Glacial Maximum (19-26 ka),
precipitation in western and southwestern USA has been shown to be ~0.75–1.5 and ~1.3-1.6 of modern, respectively (Ibarra
et al., 2018). Additionally, GCM models show a general increase in precipitation in the study area during the late Pleistocene
(Roberts et al., 2014 (Fig. 2); McGee et al., 2018 (Fig. 2)). Our observations agree with other workers who interpreted the
changes in channel size as an increase in mean discharge during this period (Alford and Holmes, 1985; Gagliano and Thom,
1967; Saucier and Fleetwood, 1970; Sylvia and Galloway, 2006). Observations of larger paleo-channels during this period are
also seen across rivers in the southeast of Texas (e.g. Bernard, 1950; Blum et al., 1995; Sylvia and Galloway, 2006), Arkansas
and Louisiana (e.g. Saucier and Fleetwood, 1970), and Georgia and South Carolina (e.g. Leigh and Feeney, 1995; Leigh et al.,
2004; Leigh, 2008).
Our contribution to the existing work on terraces in this region is to reconcile the literature that suggests an episodic
cut and fill and/or base level change model (Blum et al., 1995) with the literature on terrace formation due to increased
discharge (Sylvia and Galloway, 2006). While both have the potential to generate terraces, intrinsic processes such as bend
cut-off and unsteady lateral migration during constant base level fall need to first be ruled out. For example, relatively slow
vertical incision rates especially, pulsed discharge changes (allogenic process) and unsteady lateral migration (autogenic
process) showed indistinguishable morphologies in Limaye and Lamb (2016). Here, for all but the high Deweyville, autogenic
triggers for terrace development cannot be ruled out.
The results presented here demonstrate that it is critical to understand the many potential forcings (both allogenic and
autogenic) on a river system that can lead to terrace formation and to employ robust, quantitative tests for discriminating
between these forcings before using terraces to reconstruct paleo-environmental histories. The method proposed here for
assessing the role of allogenic processes in terrace formation using the variability of terrace elevations provides a simple,
quantitative test, and may prove useful for interpreting terrace formation in other river systems. We were not able to correlate
terrace levels back to distinct trigger events, although allogenic forcings such as sea-level fluctuations and discharge changes
were also classified here. We suggest that paleo-channel characteristics are a more faithful record of discharge changes in
fluvial systems and that additional bend metrics can differentiate autogenic terrace formation processes, specifically bend cut-
off from unsteady lateral migration rates.

## Data availability

The lidar dataset was acquired from the Texas Natural Resource Information System (TNRIS) at https://tnris.org. Please see the reference to each dataset. Tables with analysis produced from the lidar datasets are included in the supplementary material.

## Author contribution

All authors designed the analysis and contributed to the manuscript writing. HHG and TE analyzed the lidar dataset. TG developed the code to fit planes to the lidar dataset.

## Competing interests

The authors declare that they have no conflict of interest.

## Acknowledgments

We thank Webster Mangham for providing the report from the Trinity River Authority of Texas Phase II, Bathymetry and Sediment Collection for the Port of Liberty Study. We thank Ajay Limaye and one anonymous reviewer and Ajay Limaye for constructive reviews. We also thank Joel Johnson and Gary Kocurek for insightful comments on early versions of the manuscript as well as Niels Hovius, Mike Lamb, Paola Passalacqua, and Daniella Rempe for additional comments. HHG was funded by the Jackson School of Geoscience Recruiting Fellowship and the National Science Foundation Graduate Research Fellowship.

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
