# Peer review of "A multi-proxy assessment of terrace formation in the lower Trinity River valley, Texas"

_Earth Surface Dynamics, 2021_

## Author Response (AR1)

We thank the two reviewers for the constructive and thoughtful feedback. Addressing the comments and suggestions resulted in a clearer manuscript and stronger discussion. The reviewer feedback is listed in black with the response below in red.

**Reviewer #1**

**Major points**

1. The approach presented in this study largely depends on the terrace-elevation dataset, derived from a 1m resolution lidar DEM. For each of the 52 preserved terrace segments, the authors extract the median elevation and the according interquartile range (line 134, Fig. 3A). In a second step, the authors calculate the detrended elevation for each segment, by subtracting the elevation from plane fitted to the modern valley floor elevation (extracted from a 10 m resolution DEM). This approach raises several questions.
    a. First, from my understanding, this approach assumes that the elevation of a single terrace above the modern channel is constant along the valley. However, this is not necessarily the case, especially not when terraces were formed due to changes in upstream discharge and/ or sediment supply (e.g., Tofelde et al. 2019; Poisson and Avouac 2004). In such cases, the elevation of a single terrace above the river decreases in downstream direction. Because terrace elevation above the modern river can vary through space and terrace width might vary along the valley, the approach of representing each terrace only by a single (median) value might introduces biases, especially since individual terrace segments are up to 10 km in length (lines 278 and 279).

The median detrended elevations were calculated by averaging the difference in elevations for each terrace point and its respective point on the modern valley plane at the DEM resolution. The main purpose of this operation was to create a stratigraphic positioning for each terrace regardless of location along the valley (Fig. 8). These median detrended elevations were calculated from the detrended 5m resolution DEM, which allowed us to plot the detrended elevation histograms (Fig. 4). We now realize that the wording in the manuscript was misleading and have updated the manuscript to clarify how the data was processed and its purpose.

    b. Second, the detrending is done using a fitted plane to modern channel elevation data. Why was the modern elevation data extracted from a DEM with a 10m horizontal resolution and not from the 1m DEM?

From our initial inspection of the data, it was clear that the valley bottom would be best fit by a planar surface. This is defined by 4.5 million points, more than enough to constrain the fitted surface. The 10-m sampling resolution was still finer than the major topographic variability produced by the Trinity River channel. In addition, a 10m resolution dataset allowed for a reasonably fast plane fitting analysis.

    c. In general, most longitudinal river profiles are characterized by a concave-up shape. Therefore, I am wondering why the authors decide to fit a non-curved

plane to the modern channel along a ~ 40 km channel reach?

We agree with the reviewer that many longitudinal profiles are concave up. Many river segments are also linear. Our experience is that even very long sections of sand-bed rivers are best fit by linear profiles. Several factors contribute to this including that the number of entering tributaries is low and there is minimal change in both channel width and grain size. In addition, a linear profile is consistent with a large "diffusivity" or transport coefficient expected of most sand-bed rivers. An inspection of the topographic data reveals the long profile is mostly linear with slight devation in the most downstream 10km. See, for example, updated Figure 3A.

      d. What is the RMSE of the plane fitted to the modern channel elevation and how do the according residuals look like?

The RMSE of the modern valley is 1.36 m and the residuals in the N-S transect are plotted in Figure 3A.

      e. Wouldn't it be better to fit several planes to individual segments along the river in order to capture any potential curvature? The authors indicate that the modern channel slope is roughly constant along this reach (lines 173-174).

We thank the reviewer for this suggestion and agree that we want to find the most representative surface for the valley floor. A single plane fits the surface well. Meanwhile, using multiple planes would introduce errors at their joins that are difficult to correct. Finally, since our terrace plane fitting is linear and along a similar reach as the modern valley plane fitting, we wanted to keep a consistent plane-fitting methodology between the terrace and modern valley plane fitting.

      f. However, figure 3 and in particular figure 4 show that a fair amount of datapoints (~half of the lowest terraces) end up with a negative detrended elevation. Negative elevations are not possible and a median detrended elevation for the low Deweyville terraces of 3 cm (line 141) seems very unrealistic. I am not familiar with the study area, but I can't image that the previous studies (Blum et al. 1995; Garvin, 2008) have mapped river segments of less than 3 cm above the modern river as terraces.

We thank the reviewer for pointing out what might at first appear to be an inconsistency in the detrended terrace elevation data. The text has been expanded to clearly address this concern. The modern valley floor is presently net depositional and the surface has been aggrading during the late Holocene. As a result, low Deweyville terraces located away from the Trinity River and this deposition, occupy particularly low spots on the valley floor and have slightly negative detrended elevations associated with them. Partially to complete burial of low Deweyville terraces by deposition associated with Holocene sea-level rise has been described by several works including Blum et al., (1995) and Blum and Aslan, (2006).

      g. Taken together, I am wondering if the presented approach is actually able to capture terrace levels of a few meter elevation difference, given the uncertainty of the method itself? I suggest to add more details on the quality of the fitted plane to the modern channel (e.g., RMSE, residuals, etc.) and

discuss limitations of the methods including the negative detrended values. If the error of the methods itself is on the order of a couple of meters, can it still be used to distinguish terrace levels separated only by a couple of meters?

We have added the necessary analyses to document the accuracy and precision of the chosen modern valley plane fitting methods. The method focused on the terrace plane fitting is independent of the valley fit and the strength lies in determining terrace formations in low-sloping environments that might only be separated on the order of meters.

2.
    a. The second main dataset is the reconstruction of paleo discharge from preserved channel geometries on terrace treads and grain size data. I think this is a reasonable approach that was performed carefully. The results indicate that water discharge was about twice as high during the formation of the lower and middle terraces compared to modern times and times when the highest terraces were formed (Fig. 8, lines 228, 233-235, 304-308). Following theory (e.g., Parker 1979; Wickert and Schildgen 2019), a doubling in water discharge (while keeping all other parameter constant) will result roughly in a channel gradient reduced to half, triggering incision and potential terrace formation.

While we agree that "keeping everything else constant" would approximately halve the slope with a doubling of discharge, we have direct observations that indicate everything else did not remain constant. The preserved terraces document a constancy of slope even while water discharge is changing. A constant slope is predicted by the theory of Parker et al. (1998) if the increase in water discharge is matched by a comparable increase in sediment discharge so that their ratio remains approximately constant. Constant long profiles through time seem to require that overall incision is a base level signal associated with sea-level fall as shown in the experiments of Tofelde et al. (2019).

    b. Therefore, I do not follow the conclusion that the highest terrace was formed due to an external forcing, while the lower most and potentially the middle terrace level were not. I would argue that the snapshot in time preserved in the terrace surfaces is the switch from aggradation or stable bed elevation to incision. If the lower two terrace levels indicate particularly wet conditions, isn't this rather indicating that those terraces were formed due to an allogenic forcing (= increase in discharge), while dry conditions during the formation of the abandonment of the highest terrace argue against discharge increase as a driver for those oldest terraces?

The reviewer's comments have been extremely helpful in guiding us on how to improve our arguments on testing allogenic versus autogenic terrace formation in the text. We will summarize these arguments here. First, terraces produced via channel-bend cutoff and abandonment under conditions of overall incision (regardless of the cause of this incision) are referred to as autogenic because they are directly connected to properties of channel kinematics and not a system-scale change external to the river. Terraces connected to ongoing channel kinematics are not expected to be well organized in space, while a set of allogenic terraces simultaneously developing in response to a system scale perturbation are expected to be well organized in space. Even though we are able to place quantitative

constraints on a significant discharge change through time, the observed constant bed slope indicates this environmental change is not driving incision and does not contribute to allogenic terraces. The allogenic forcing for all observed terraces is interpreted as sea-level fall and only the oldest, highest terraces seem to preserve a record of a sea-level fall event and are therefore considered allogenic. All other terraces preserve a signal of channel kinematics, which are autogenic.

3.

    a. Taken together, I think the presented approach is interesting. But due missing information on the reliability of the elevation dataset and the reconstruction of wettest conditions during the formation of the lower two terrace levels, I currently can't follow the conclusion of the authors (e.g., lines 17-18) that only the uppermost terraces were formed by an allogenic mechanism and the lower terraces by an autogenic mechanism. As there is no independent dataset to validate the conclusion drawn in this study and the conclusion also differs from previous work (lines 56-59), I think it would be useful to discuss those discrepancies and provide arguments for why the conclusion presented here is more reliable than the previously proposed mechanism.

We have clarified our approach for terrace elevation sets and the associated accuracy of the lidar and the fits of the plane to the surfaces. Thank you for the feedback and this opportunity. We recognize that the main goal of this study is to determine if an autogenic terrace process can be ruled out and provide statistical methods to distinguish allogeneic from autogenically formed terraces. Autogenic terrace formation has not been previously evaluated for Deweyville terraces. We have expanded our discussion to elaborate on why autogenic terrace formation should be considered as the null hypothesis.

**Minor points**

4.

    a. Introduction: In my opinion the introduction is in parts incomplete. The main goal of the study is to distinguish between autogenic and allogenic terrace formation from topographic analyses. As such, the introduction should contain a complete description of the different terrace formation mechanisms and the resulting topographic differences. Lines 38 to 45 briefly explain the two general mechanisms of allogenic forcings – downstream variability in base level elevation and upstream variability in the water-to sediment ratio. And lines 51 to 54 briefly list some, but not all the mechanism of autogenic terrace formation. For example, the complex response/internal feedback mechanism described by Schumm (1973) and further studies (Slingerland and Snow 1988; Stanley A Schumm and Parker 1973) could be added here.

The introduction has been improved and now includes a more thorough explanation of terrace formation processes. We agree with the reviewer that our review of autogenic processes is not exhaustive, rather focuses on those relevant to the Trinity system. The papers suggested by the reviewer specifically focus on changes in slope over time, as well as changes in channel pattern and sinuosity. Records of these changes are not preserved in this system and are therefore not discussed here.

    b. Later in the manuscript, the authors investigate the number of preserved meander bends on each terrace. If this is a topographic indicator of terrace

formation, I suggest to introduce it as well in the beginning of the manuscript. Same for paired versus unpaired terraces, which is only introduced as an indicator during the discussion (lines 317-318).

We have added discussion of preserved meander bends, expected elevation drop, and paired vs unpaired terraces to the introduction.

  c. So, what I miss in the introduction is a comprehensive description of driver-dependent topographic differences in the resulting terraces. What are expected differences, for example, in terrace surface slope and according differences in terrace elevation above the modern channel, channel width, terrace surface age, the predominant formation of paired or unpaired terraces, number of preserved bends, average size of terrace segements etc. for each mechanism of terrace formation (upstream allogenic, downstream allogenic, autogenic). For example, Poisson and Avouac (2004) investigated terrace in the Tien Shan formed by an increase in water discharge through time. The resulting terraces decrease in gradient, resulting in higher elevation differences between terraces and the modern channel at the upstream end. Tofelde et al. (2019) have investigated topographic differences in terrace and channel geometries in response to upstream and downstream perturbations using lab experiments. Those experiments also show a decrease in river channel gradient relative to terrace surface gradient for terraces formed by a decrease in the sediment-to-water discharge ratio. Malatesta et al. (2017) have modeled autogenic terrace formation numerically, and showed that those terraces were mostly unpaired. These are only some examples, but I think introducing all investigated parameters and summarizing expected differences caused by autogenic and allogenic drivers will help to later argue for or against one or the other driving mechanism.

The introduction has been updated to include the suggested references and description of driver-dependent topography differences. We agree that these additions improve the introduction and the setup for interpreting the data from this specific system.

  5.
    a. Data and methods & Results: My impression was that some of the results were presented in the Data and Methods section (e.g., lines 258-259, 262, 265-267) and other results in the Results section. When reading it felt like I was going through the same plots twice. I think this mainly results from the fact that most figures showing results are already referenced in the Methods section and some of them again in the Results section. Figure 10 (elevation difference between terraces and number of bends), in contrast, is only presented in the methods section, but not in the Result section.

A similar suggestion was made by Reviewer #2, who suggested relabeling the results as "synthesis of observations". We have also relabeled the data and methods section to "approaches and observations" to reflect the results included in this section.

    b. I suggest to reduce the Data and methods section to a brief description of all applied methods (without any references to the figures), and concentrate the description of all datasets in the Results section.

We appreciate the reviewer feedback. See comment 5a for how we addressed this.

6. Figure 1&2: I would find a regional map in addition to the map of the study site very helpful. Currently it is hard to estimate, for example, the distance to the ocean. Also, figure 1 and 2 miss coordinates, which makes it complicated to find the study site for example in google earth.

An updated figure has been added to include the suggestions above and it includes estimates of the shoreline during terrace formation.

**Line-by-line**

Lines 14-16: I would expect a clustering of terrace elevation for some allogenic drivers, but not for all (e.g. not necessarily for changes in water-to-sediment discharge ratios); see comment above.

We have addressed this comment in our response to 2b above.

Lines 126-127: This long list of numbers is a little confusing. In which order are the numbers given? For clarity, I suggest to separate them in two or four lists, respectively.

We have separated the list into two to describe the horizontal and vertical accuracy.

Line 133: Why was a 5m grid used here and not the higher resolution 1m grid?

We used a 5m grid because that's the size of a dataset that could be easily loaded into Matlab given the computational resources readily available. This justification has been added to the text.

Line 140: Comma between 'elevations' and 'is plotted…'?

We have added the suggestion.

Lines 158-162: But don't autogenically formed terrace also preserve the slope? Or is the point to stress that autogencially formed terraces are on average shorter than allogenically formed ones? If so, it might be interesting to include a plot showing the size distribution of the different terrace segments for each level?

We agree that autogenically formed terraces preserve local slopes. Here we are highlighting that a set of autogenically formed terraces are less likely to preserve a paleo-valley plane since they will have been abandoned at different times.

Lines 184-185: This sentence is confusing. I suggest rephrasing.

We have incorporated the suggestion.

Line 221: A legend could be added indicating which datapoint size relates to what terrace length.

We have changed the figure to include the color scale highlighting the size of

paleo-channel length.

Line 248-249: Remove 'decrease in…'?

We have made the suggested changes.

Lines 279-281: I suggest to introduce this parameter (paired vs. unpaired terraces) already in the introduction.

We have added the suggested information in the introduction.

Lines 322-325: Also, the number of channel bends used as an argument to distinguish between allogenic and autogenic drivers could be introduced earlier, e.g. in the introduction.

We have added the suggested information in the introduction.

Lines 257-258: The sentence sounds strange, I suggest rephrasing.

We have rephrased the highlighted sentence.

Lines 383-384: If those terraces are formed by meander-bend cut-off would they expected to be up to 10 km long? This again makes me think that a plot showing the distribution of terraces lengths might be helpful.

We have plotted the length and width of terraces to show their distribution in the figure below.

**Reviewer #2 Ajay Limaye**

1. Hassenruck-Gudipati et al. use morphometric analyses of alluvial terraces and associated paleochannels along the Trinity River, Texas, to test end-member hypotheses for river terrace formation. These end-member scenarios are (1) autogenic terrace formation and (2) terrace formation driven by episodic incision caused by sea-level change. The analysis informs the interpretation that the terraces record past base-level fall (for the highest terrace) and autogenic processes (for lower terraces). The study treats a timely question and applies creative approaches to the data analysis, including Monte Carlo simulation to interpret both the terrace and paleochannel datasets. The study site is well justified based on prior terrace studies and paleodischarge reconstructions in the Texas Coastal Plain region.

2. I have three main comments. First, the geologic context for the analysis could be further developed. For readers without extensive prior knowledge of the studies on the Colorado and Brazos Rivers, it may be difficult to place the new analyses in perspective. I made several suggestions in the line-by-line comments to address this gap, including adding an area map, an alluvial stratigraphic column, and a sea-level curve – any of which would help to evaluate and contextualize the interpretations in this study.

We agree with both reviewers that additional information providing geologic and geographic context is needed. Because of this, we have added the suggested map, stratigraphic column, and sea-level curve to the document.

3.
   a. Second, the manuscript focuses on two terrace formation pathways; however, the analysis points to third important factor, namely, large-magnitude changes in discharge (L353-370). The implications of this finding can be further developed.

This is a fair point by the reviewer, although we may respectfully disagree that a large change to water discharge was a factor in terrace formation. As already discussed while addressing comments by Reviewer 1, the changes in water discharge seem to have been balanced by changes to sediment discharge so that slope or long profiles remain approximately constant over time. Changes in slope are required for discharge variation to be considered a major terrace-formation pathway and these are missing from the record. This point is now discussed in the text so we appreciate the suggestion to expand on the implications of these findings, and have done so in the revised discussion. The remainder of this reviewer's comment is addressed below in separate parts (for clarity).

   b. If sediment discharge increased at the same time to maintain a consistent channel slope (as proposed in L375-376), can you test this hypothesis using other sedimentary data?

To address this point we summarise previous estimates of temporal change in sediment discharge through the Trinity system and present this data in the Discussion. Theory predicts that constancy of slope of under changing water discharge requires comparable adjustments in sediment discharge so that their ratio remains approximately constant (e.g., Parker et al., 1998). Observations made during this study should motivate future investigation to look at total sediment flux via bedform analysis as suggested by Mahon and McElroy (2018), as well as other methods.

   c. Moreover, the paleodischarge reconstruction, which indicates much larger discharges during the formation of the intermediate Deweyville terraces, also seems to complicate the story of the intermediate terraces being autogenic. What does it mean for the terraces to be autogenic if they formed during a period of significantly higher water discharge? Does it mean that terraces in general are not faithful proxies for paleoclimate? Squaring the terrace interpretations with the paleodischarge interpretations would be a very helpful contribution, and based on these data, might even push us to refine what we mean by "autogenic" terraces.

We agree that the occurrence of a large temporal change in water discharge during the terrace-formation interval needs expanded discussion. This new text is found in the Discussion section. To summarize our thoughts here, changes in water discharge that are balanced by commensurate changes in sediment discharge so that river slopes remain constant are not viewed as an important driver in terrace formation, while systems in which slopes change through time because water and sediment are not matched, can produce allogenic incision and more obviously fit into a terrace-formation analysis. Since the flow of both water and sediment must be considered to accurately predict changes to the slopes of river systems, terrace studies need to consider both to have any hope of providing an

accurate paleoclimate interpretation.

In the case of the Trinity, the terraces are not faithful recorders of paleoclimate change since there is no distinguishable recorder within the elevation data. However, the paleo-channels themselves clearly show the paleoclimate variations.

4. Third, the terrace analysis centers on the hypothesis that the variance in elevations on a terrace results from formation mechanism (i.e., autogenic versus allogenic). However, there is a significant potential source of variance in the terrace elevations that is not sufficiently described: dissection of the terraces after formation. For example, Figure 7D shows a gully eroded into terrace T6, which presumably increases the variability in elevation on that terrace surface. Does the analysis account for or exclude this additional source of elevation variance that is unrelated to the main hypothesis? Further explanation of the mapping methods might address this point.

The reviewer makes an important observation that terrace elevations can be altered by dissection and deposition of gullies and their sediment. We have added additional information to our mapping approach of avoiding including those areas in elevation maps and have added the terrace extents used in our elevation variance analysis to Figure 7 to highlight this.

5. Overall, this is an ambitious and exciting study. With revisions to deepen the geologic context, reconciliation of the competing hypotheses, and justification for the mapping methods, I think this work can make an important contribution to current debates regarding alluvial rivers and paleoenvironmental reconstruction.

We thank the reviewer for the detailed and constructive feedback.

**Line-by-line comments:**

L25-27: "often host remnant river-channel segments": The paleochannels for the Trinity River terraces are exceptionally well expressed compared to many other locations. Add this context, or references to indicate how common these types of paleochannels are.

Revised text to reflect that not all terraces host exceptionally well-expressed paleochannels.

L27-29: "terrace formation requires…": other mechanisms include alluviation and incision caused by valley damming (e.g., Mackey et al., 2011, doi: 10.1073/pnas.1110445108), land-use change (Womack and Schumm, 1977, doi:10.1130/0091-7613(1977)5%3C72:TODCNC%3E2.0.CO;2), and perhaps others. Consider using a more general definition for the required conditions to form a terrace.

The references and mechanisms have been added to allogenic change driven by sediment to discharge changes.

L36: Which coastal deposits, specifically?

We specified the deposit as deltaic.

L63: "Allogroup": Define what this is or replace with a more widely known term.

We defined Allogroup in the text.

L64-66: As a foundation for these hypotheses, introduce the observation that the three sets of terraces occur at different elevations. L55-56 could be a good place.

We have added a sentence clarifying that terrace sets have been interpreted to exist at three different elevation levels.

L73: Check grammar, currently reads as if the hydrograph is prone to flooding.

We changed the text to correct this.

Figure 1: Beautiful data, might look even better with some adjustments to the figure layout. Placing the legend between the two maps distracts somewhat form the maps. One alternative idea is to rotate the maps so that they go left-to-right, then place the annotations on the side.

We have updated the figure to reflect these suggestions.

Figure 1 caption: The (A) and (B) labels seem to be misplaced. More common to put the label ahead of the thing it describes.

We changed the text to refer to the labels before the description.

Figure 2: color scale lacks units. Is this detrended or absolute elevation?

This is absolute elevation and the legend has been updated to reflect this.

L84-103: The stratigraphic information is useful, but hard to digest for the uninitiated. I suggest adding a stratigraphic column to tie the work to the existing alluvial stratigraphic framework that is so well developed in this region.

A stratigraphic column has been added to Figure 2.

L104-105: The sea-level history is central to evaluating the main claim in this manuscript, i.e., that some Trinity River terraces reflect base sea-level change while others do not. As with the stratigraphic information, it would be very helpful to show the sea-level reconstructions.

A global sea-level reconstruction has been added to Figure 2.

L113-121: Several of the referenced studies treat rivers in the Texas Coastal Plain, such as the Colorado River. However, none of these rivers are named in the Introduction and there is limited information to place these studies in spatial context. An overview map that includes the Trinity River and other major rivers in the region would help.

We have added rivers with previous Deweyville studies to a regional overview map.

L126: More common to report grid spacing (i.e., 1 m).

We changed the text to reflect this.

L130-131: Did these previous studies map the terraces, and this study is refining those maps using high-resolution topography data? The relationship between the existing maps and this work is somewhat unclear.

We clarified the text to indicate that previous studies mapped the terraces on the Trinity using a combination of satellite and DEM data (resolution unknown). We adapted the terrace mapping with expectations of breaking up terraces if we didn't believe they were genetically connected. Additional terraces we identified along the valley wall were also mapped.

L133: State the rationale for sampling the data at this lower resolution.

We have added a rationale in the manuscript.

L135: Strictly speaking, latitude and UTM northing (Fig. 3A) are not equivalent.

We have updated the wording of the manuscript to reflect this.

L135-136: Is the plane-fitting procedure suitable for these data? Figure 3A shows the trend line from the fit but not the underlying elevation data.

We have updated the figure to include the residual of the underlying elevation data.

Figure 3: What do the error bars in Fig. 3A represent? Also, the readability for the x-axes in both plots could be improved by plotting as distance in kilometers.

We have updated the caption for Figure 3 and changed the x-axes to kilometers.

Figure 3 caption: same comment regarding labels as for Figure 1.

We have updated the labeling of Figure 3.

L155: Section 3.1: This section expresses the core hypothesis. Consider moving this section before the data in L122-154.

We have moved this section to a location earlier in the manuscript.

L163-164: Does this hypothesis consider the Finnegan and Dietrich (2011) model for terrace abandonment driven by autogenic knickpoints? It seems plausible that such knickpoints would abandon terraces with low RMSE in their plane fits.

We agree that individual terraces with an autogenic knickpoint terrace abandonment would have low RMSE. However, a set of terraces is likely to have RMSE values that exceed individual RMSE values for this mechanism.

L173: As noted above, the valley profile is not actually shown, only the linear fit to the

valley profile.

We have updated the figure to include this.

Figure 3 caption: The meaning of the colored line in each plot is unclear. Does this represent the random terrace grouping, and if so, how?

The colored lines in Figure 6 refer to the RMSE values of the plane fitting associated with the terraces originally assigned to each group. We have updated the caption to clarify this.

L196: Note in the text that this is the sand-bed (rather than gravel-bed) version.

We have updated the text to reflect this.

L234-235: These modern discharge statistics are important context for the paleodischarge estimates. Can you add the modern statistics to Figure 8?

We have added modern river bankfull value to the figure.

L237: "To further test the statistical groupings within our terrace and paleo-channel data": lost the thread regarding the specific purpose of these statistical tests. Are you testing for the existence of groupings, the number of groupings, or something else? How does this analysis contribute to testing the main hypothesis?

We are testing the existence of groupings here. The text now notes that results are used to assess if it's appropriate to divide Deweyville terraces into three distinct sets.

L264-265: The relevance of the number of paleochannel bends in the terrace is unclear. This hypothesized link is stated later, however, it's unexplained at this point in the text.

We have moved the hypothesized link before this line.

L275: 10 figures and many results have already been presented before arriving at section 4, "Results." Based on the content, it may be more fitting to rename this section along the lines of "synthesis of observations."

We have updated the section to the suggested name.

L316-323: This passage can be refined to better capture some specific points in the referenced studies. Greater precision is needed to specify which autogenic process is being discussed and to explain how that process would impact the terrace characteristics such as pairing, age, and consistency of elevations.

 The model results in Limaye and Lamb (2016) indicated that a specific autogenic process – a meandering river undergoing constant vertical incision – can make terrace features typically interpreted to reflect external (allogenic) processes. Specifically, relatively low vertical incision rates (~ 0.1 mm/yr) can yield terraces that appear to be paired and are relatively long (> 10 channel widths). The current text does not capture these points. Also, in L325 only the Finnegan and Dietrich (2011) paper treated enhanced erosion rates driven

by cutoffs.

We have updated the text throughout the paper to differentiate between autogenic processes.

L342: More specificity about the autogenic process(es) considered would be helpful.

We have updated the text to be more specific.

L352: What does your new analysis mean for interpreting the geochronology data summarized in the Introduction (ca. L100)?

We have added the implications of our new analysis on interpreting geochronology data.

L355: *Phillips

We have updated the text.

L381-384: As above, a more robust argument is needed to relate the number of bends preserved to the terrace formation mechanism, as relatively long terraces have been proposed to form autogenically.

We have updated the text to specifically highlight the link between the number of bends and the bend cutoff autogenic trigger for terrace formation.

Supplementary files: It would be helpful to include a readme file to explain the purpose of each file, data formats, conventions, etc.

We have updated the supplementary files to include a readme.

---

## Referee Report (RR1)

**Review of *A multi-proxy assessment of terrace formation in the lower Trinity River valley, Texas***

I have reread the manuscript *A multi-proxy assessment of terrace formation in the lower Trinity River valley, Texas* by Hassenruck-Gudipati and colleagues and their responses to the two previous reviews. The authors addressed most of the comments raised in the two reviews and implemented them in their revised manuscript. The revised manuscript reads well, is much clearer and well structured, and most concerns have been addressed. I think that the manuscript will be an important and timely contribution to the community. But there is one important point that I raised during the first review, which is still not entirely clear to me. Therefore, I need to address it again and I suggest to clarify this prior to publication.

The authors propose to use variability in terrace heights as a test to assess the plausibility of an allogenic terrace formation mechanism. To do this, they (1) subtract terrace heights from a plane fitted to modern floodplain heights (Figs. 3 and 4), and (2) compare the RMSE of terrace heights relative to a plane fitted to all data points on the same terrace with the RMSE of a plane fitted to randomly selected terrace segments, which is an indicator of autogenic terrace formation (Figs. 5 and 6). If I understand correctly, this is to investigate whether all terrace segments of a terrace (low, medium, high) are similar in height and belong to one large, externally-driven incision event or whether the heights are scattered and the terraces were formed by individual, localized incisions (autogenous terrace formation). However, using a plane as a reference surface introduces some uncertainty in the data, which I have tried to outline in the figure below. Although the modern river profile is fairly straight, elevation values are above the trend line near the outlet (probably due to recent sea-level rise and corresponding sediment deposition), below the trend line in the middle part, and above the trend line again in the upper part (Fig. b, taken from the manuscript). This variability in modern floodplain elevations results in an overall RMSE of 1.36. However, there is a spatial trend in the residuals that will also affect the detrending of the terrace elevation data. In the schematic figure on the left (a), the offset between the terrace surface and the modern floodplain is constant along the channel, as assumed for an allogenic forcing such as a base-level drop. However, the chosen approach systematically results in lower detrended values for the terraces in the middle of the reach compared to the upstream and downstream ends (blue lines). To me, this means that any distribution of residuals in the terrace data that results in an RMSE on the order of 1.36 is entirely due to the method itself. The RMSEs for the three terrace data sets are only slightly higher (1.43 m, 1.54 m, and 1.41 m). Is it possible, then, that most of the scatter in the terrace data is caused by the method, while only a fraction is truly due to variability in terrace heights?

[Figure]

The authors then compare the RMSE, which describes the offset between each terrace height and a best-fit plane, to the RMSEs of only randomly selected terrace segments (Fig. 6) to test the null hypothesis. Figure 6 shows that the overall RMSE for randomly selected terrace segments increases with the number of segments selected, which I would expect since a larger number of randomly selected terrace segments causes a wider distribution of residuals. Currently, the authors do not reject the null hypothesis for the lower terraces because the RMSE of the lower terrace overlaps with the RMSE distribution of the Monte Carlo fits (Fig. 6). However, the non-rejection is not due to the fact that the elevation data of the lower terrace has a larger dispersion compared to the other terraces (given the RMSE, it is quite similar to the middle and high terrace levels), but because fewer segments are preserved. Does this mean that even if the distribution of the residuals and the RMSE are very similar, the question of whether a terrace group can be considered allogenically formed or not depends solely on the number of preserved terrace segments?

In any case, an RMSE is only a single parameter describing a distribution of residuals. Wouldn't it therefore make more sense to compare the full distributions of residuals to assess the scatter in the terrace survey data, perhaps using a Kolmogorov-Smirnov test? After all, as long as the distributions of the residuals are quite similar to the modern flow, an allogenic driver seems quite reasonable.

On the other hand, I understand that the authors prefer to test the null hypothesis of an autogenic driving mechanism. The current overlap of the RMSE values of the lowest terrace and the randomly selected terrace segments cannot falsify this hypothesis. However, this means that there are not enough segments left to identify an allogenic drive. This does not mean that these terraces were autogenically generated. It just means that not enough segments are preserved to determine this. In this case, I cannot support the conclusion that the lower terrace was formed by an autogenic mechanism, as stated in the abstract (line 18) and in several places in the manuscript.

Overall, I think the manuscript is an important contribution to the community and that we lack methods to serve as a "quality control" before using terraces for paleoenvironmental reconstructions. But I am not yet fully convinced that the proposed approach or the conclusion drawn are correct. I also realize that the elevation data are only one of several proxies analyzed. However, given the exceptional preservation of the paleochannels at the study site that were used for the other proxies, the analysis of the elevation data is the one that can be most easily applied to other study sites. Therefore, I would be grateful if the point raised above could be clarified before publication. I provide some further line-by-line comments below.

**Line-by-line comments**

Lines 15-16: A cluster in elevations is not necessarily expected for terrace formed by a change in hydroclimate, as is also explained well later in the manuscript.

Lines 45-46: To make the sentence easier to read, it might helpful to add a "(1)" before 'punctuated decreases' and a "(2)" before 'punctuated base-level fall'.

Line 62: 'This reduction from the measured paleo-slopes of terrace sets…' This sounds a little strange, I suggest rewording.

Line 77: Channel bed slope instead of bedrock slope? (last word in line)

Figure 1: Maybe add an arrow indicating the flow direction in the figure. Also, is it correct that river discharge decreases in downstream direction? The downstream gauging station (Liberty) has a lower discharge value compared to the upstream one.

Figure 2: Please add coordinates to the map (A) to allow the reader to find the site in other datasets and Google Earth. Is the legend in (B) displayed correctly? To me the colors for post-Deweyville, Beaumont and Lissie all look white.

Lines 131-132: The information about floodplain aggradation during the Holocene is an important point. It means that we cannot directly compare the slope of the valley floor with the slope of the terraces, because the valley floor slope at the end of incision phase is not preserved anymore. Hence, this argument cannot be used to rule out hydroclimatic changes as terrace formation drivers, because it is possible that the channel slope at the end of the incision phase was different than the terraces. Instead, the similarity in slopes of the three terrace themselves could be used as an indicator that any potential changes in water discharge were complemented by changed in sediment discharge.

Line 157: The summary of the null hypothesis and approach in section 3 is really helpful. Just a suggestion, but the authors could even consider to summarize their approach in a simplified, schematic sketch, especially since they want to 'sell' this approach for future studies.

Line 187: It is unclear if the median values were calculated for each of the 52 terrace segments or only for the 3 terraces. Please clarify.

Line 191: As stated above, the higher elevation values close to the outlet that plot above the plane are probably related to sediment deposition since sea-level rise?

Line 192-194: I suggest to move this sentence up to line 189 to state from the beginning, why this analysis is done.

Figure 3: It would be helpful to color the datapoints in (A) and (B) according to the terrace they belong to.

Figure 6: The actual RMSE values for the three terraces are not give here, they only come up later in section 4. Please briefly give the values already when describing the fits in the results.

Line 506: Remove 'introduce'?

---

## Author Response (AR2)

**Review of *A multi-proxy assessment of terrace formation in the lower Trinity River valley, Texas***

I have reread the manuscript *A multi-proxy assessment of terrace formation in the lower Trinity River valley, Texas* by Hassenruck-Gudipati and colleagues and their responses to the two previous reviews. The authors addressed most of the comments raised in the two reviews and implemented them in their revised manuscript. The revised manuscript reads well, is much clearer and well structured, and most concerns have been addressed. I think that the manuscript will be an important and timely contribution to the community. But there is one important point that I raised during the first review, which is still not entirely clear to me. Therefore, I need to address it again and I suggest to clarify this prior to publication.

We thank the reviewer for their feedback on the updated manuscript and highlighting outstanding concerns. We address these below and in the updated the manuscript.

The authors propose to use variability in terrace heights as a test to assess the plausibility of an allogenic terrace formation mechanism. To do this, they (1) subtract terrace heights from a plane fitted to modern floodplain heights (Figs. 3 and 4), and (2) compare the RMSE of terrace heights relative to a plane fitted to all data points on the same terrace with the RMSE of a plane fitted to randomly selected terrace segments, which is an indicator of autogenic terrace formation (Figs. 5 and 6). If I understand correctly, this is to investigate whether all terrace segments of a terrace (low, medium, high) are similar in height and belong to one large, externally-driven incision event or whether the heights are scattered and the terraces were formed by individual, localized incisions (autogenous terrace formation). However, using a plane as a reference surface introduces some uncertainty in the data, which I have tried to outline in the figure below. Although the modern river profile is fairly straight, elevation values are above the trend line near the outlet (probably due to recent sea-level rise and corresponding sediment deposition), below the trend line in the middle part, and above the trend line again in the upper part (Fig. b, taken from the manuscript). This variability in modern floodplain elevations results in an overall RMSE of 1.36. Is it possible, then, that most of the scatter in the terrace data is caused by the method, while only a fraction is truly due to variability in terrace heights?

We thank the reviewer for this question on what causes the variation in RMSE in the floodplain. It is true that this method does not allow to distinguish where the RMSE error is coming from. While the above is the case we do recognize that any potential systematics in the residual are what would be expected from a plane fit to a concave up longitudinal profile. However, we suspect that the slight deviations in residuals in the Figure 3 (now Figure 4) a) are due to 1) aggradation at the downstream end and 2) the river deviates from the N-S valley axis in the upstream. We also note that the actual magnitude of variability is very small, ~1-2 m over a reach of >90 km, and less than the terrace heights above the modern valley floor (see Fig. 3B).

In the schematic figure on the left (a), the offset between the terrace surface and the modern floodplain is constant along the channel, as assumed for an allogenic forcing such as a base-level drop. However, the chosen approach systematically results in lower detrended values for the terraces in the middle of the reach compared to the upstream and downstream ends (blue lines). To me, this means that any distribution of residuals in the terrace data that results in an RMSE on the order of 1.36 is entirely due to the method itself.
The RMSEs for the three terrace data sets are only slightly higher (1.43 m, 1.54 m, and 1.41 m). Is it possible, then, that most of the scatter in the terrace data is caused by the method, while only a fraction is truly due to variability in terrace heights?

We thank the reviewer for pointing out the link between assumptions of a plane fitted to modern floodplain and the resulting implications assessing the variability in terrace heights. However, we think it important to note that the plane fitting to the terrace groupings was done on the raw elevation data, not the detrended data, so there would be no propagation of error from any systematic residuals in our planar

fit to the modern valley floor to terrace analysis. The same basic concern could be raised for a planar vs. polynomial fit, but again, we argue that any systematics are within the error/natural variability of the system, and so fitting with a 2D polynomial would be an over-reach for the data.

Furthermore, terraces like those described on the Trinity River only have variations in their detrended elevation of maximum 15 m (Fig. 4). This is on the order of a channel depth for the modern Trinity River. Larger structural variation in along channel variation as describe in Figure a., below, likely only have a small effect on the RMSE results for a low-sloping river like the Trinity River. Assessing the variability in long-profile elevation versus terrace elevation variation might for rivers with larger concave up structures might be important.

[Figure]

The authors then compare the RMSE, which describes the offset between each terrace height and a best-fit plane, to the RMSEs of only randomly selected terrace segments (Fig. 6) to test the null hypothesis. Figure 6 shows that the overall RMSE for randomly selected terrace segments increases with the number of segments selected, which I would expect since a larger number of randomly selected terrace segments causes a wider distribution of residuals. Currently, the authors do not reject the null hypothesis for the lower terraces because the RMSE of the lower terrace overlaps with the RMSE distribution of the Monte Carlo fits (Fig. 6). However, the non-rejection is not due to the fact that the elevation data of the lower terrace has a larger dispersion compared to the other terraces (given the RMSE, it is quite similar to the middle and high terrace levels), but because fewer segments are preserved. Does this mean that even if the distribution of the residuals and the RMSE are very similar, the question of whether a terrace group can be considered allogenically formed or not depends solely on the number of preserved terrace segments?

We thank the reviewer for this great summary of the impacts of the number of terraces preserved on assessing if these terraces were allogenically formed. We think that it is harder to justify an allogenically formed label for the case where only a small number of terraces are formed, something our Monte Carlo approach quantifies, and which also makes sense for the reasons the reviewer points out. However, we would argue it does not "solely" depend on this, as there is nothing precluding a set of allogenic terraces from having a very low RMSE over such a short reach as studied here, the studied sets simply do not.

In any case, an RMSE is only a single parameter describing a distribution of residuals. Wouldn't it therefore make more sense to compare the full distributions of residuals to assess the scatter in the terrace survey data, perhaps using a Kolmogorov-Smirnov test? After all, as long as the distributions of the residuals are quite similar to the modern flow, an allogenic driver seems quite reasonable.

We have considered a K-S test for the residuals but given the macro-scale roughness of the floodplain and terraces, we are not convinced the residual structure would be sufficiently more informative than a simple RMSE metric. We prefer to use the RMSE metric as a simple means of evaluating the plane fit rather than the evaluating of the more floodplain structure-dependent distribution of residuals.

On the other hand, I understand that the authors prefer to test the null hypothesis of an autogenic driving mechanism. The current overlap of the RMSE values of the lowest terrace and the randomly selected terrace segments cannot falsify this hypothesis. However, this means that there are not enough segments left to identify an allogenic drive. This does not mean that these terraces were autogenically generated. It just means that not enough segments are preserved to determine this. In this case, I cannot support the conclusion that the lower terrace was formed by an autogenic mechanism, as stated in the abstract (line 18) and in several places in the manuscript.

We agree with the reviewer that the limited number of terrace segments limits the ability to reject our null hypothesis that terraces were formed autogenically. That said, we would also expect an allogenic forcing mechanism to abandon a large number of terraces considered over the same valley reach length with more than one channel bend preserved on the terraces. Aggradation history at the downstream end prevents us from assessing the first expectation but number of channel bends are less than or equal to 1 for low Deweyville terraces (Figure 12). Regardless, we wholeheartedly agree with the reviewer that our failure to reject the null hypothesis of autogenic formation for the Lower Deweyville set does not disprove allogenic formation, it simply means we should not reject autogenic formation. We have tried to make this clear in the abstract on Line 23-26. Our interpretation that favors autogenic formation of this terrace set is our interpretation based on both these results, and the rest of the analysis presented in the manuscript.

Overall, I think the manuscript is an important contribution to the community and that we lack methods to serve as a "quality control" before using terraces for paleoenvironmental reconstructions. But I am not yet fully convinced that the proposed approach or the conclusion drawn are correct. I also

realize that the elevation data are only one of several proxies analyzed. However, given the exceptional preservation of the paleochannels at the study site that were used for the other proxies, the analysis of the elevation data is the one that can be most easily applied to other study sites. Therefore, I would be grateful if the point raised above could be clarified before publication. I provide some further line-by-line comments below.

We thank the reviewer again for their comments, and hope our above response have proven satisfactory in addressing their remaining concerns.

**Line-by-line comments**

Lines 15-16: A cluster in elevations is not necessarily expected for terrace formed by a change in hydroclimate, as is also explained well later in the manuscript.

We thank the reviewer for catching this. The lines now read (L15-16): "For 52 distinct terraces, we quantify whether terrace elevations fit distinct planes…"

Lines 45-46: To make the sentence easier to read, it might helpful to add a "(1)" before 'punctuated decreases' and a "(2)" before 'punctuated base-level fall'.

We thank the reviewer for the helpful suggestion. The text now reads (L45-46): "Commonly invoked allogenic triggers connected with terrace formation are (1) punctuated decreases in sediment-to-water flux that are assumed to embed a signal of regional climate change and (2) punctuated base-level fall controlled…"

Line 62: 'This reduction from the measured paleo-slopes of terrace sets…' This sounds a little strange, I suggest rewording.

We thank the reviewer for the feedback and have rewritten the sentence to read (L62-63): "This reduction in slopes from older terrace sets to the modern floodplain has been observed in both natural (Poisson and Avouas, 2004) and experimental (Tofelde et al., 2019) systems."

Line 77: Channel bed slope instead of bedrock slope? (last word in line)

We have changed the wording to read (L77-78): "include local variations in channel dynamics, channel bed slope, and sediment contribution from tributaries."

Figure 1: Maybe add an arrow indicating the flow direction in the figure. Also, is it correct that river discharge decreases in downstream direction? The downstream gauging station (Liberty) has a lower discharge value compared to the upstream one.

We have added an arrow to identify the downstream direction. The downstream gauging station does have a lower mean discharge. This could be because not all discharges are found for USGS stage measurements for Liberty since tides do influence discharge in this region.

Figure 2: Please add coordinates to the map (A) to allow the reader to find the site in other datasets and Google Earth. Is the legend in (B) displayed correctly? To me the colors for post-Deweyville, Beaumont and Lissie all look white.

For Figure 2, we have added the coordinates to the map (A) and added colors to the legend in (B) for stratigraphic unites outside of the study period.

Lines 131-132: The information about floodplain aggradation during the Holocene is an important point. It means that we cannot directly compare the slope of the valley floor with the slope of the terraces, because the valley floor slope at the end of incision phase is not preserved anymore. Hence,

this argument cannot be used to rule out hydroclimatic changes as terrace formation drivers, because it is possible that the channel slope at the end of the incision phase was different than the terraces. Instead, the similarity in slopes of the three terrace themselves could be used as an indicator that any potential changes in water discharge were complemented by changed in sediment discharge.

We agree with the reviewer that both water and sediment discharge likely increased and point this out in the manuscript in Lines 453-455: "We suspect that the switch in discharge is not directly recorded in the terrace elevation because the change in water discharge appears to have been approximately matched by a sediment-flux increase, as recorded in the constant long-profile slope for the paleo-river. With no slope reduction, no incision would have occurred." Therefore, even if hydroclimate changed it might not have been the trigger for terrace formation. As the reviewer points out, this assessment is harder for the low Deweyville terraces that have be partially buried with Holocene deposition (L131-132).

Line 157: The summary of the null hypothesis and approach in section 3 is really helpful. Just a suggestion, but the authors could even consider to summarize their approach in a simplified, schematic sketch, especially since they want to 'sell' this approach for future studies.

We thank the reviewer for this suggestion and have summarized the approach to section 3 in Figure 3 and updated our RMSE fitting in Figure 4.

Line 187: It is unclear if the median values were calculated for each of the 52 terrace segments or only for the 3 terraces. Please clarify.

We have updated the text to specify that each of the 52 terrace segments are quantified in this way. The text now reads (L193): "From these elevations, the median value and interquartile range were found for each of the 52 mapped terraces."

Line 191: As stated above, the higher elevation values close to the outlet that plot above the plane are probably related to sediment deposition since sea-level rise?

We reemphasized this likely cause for this trend is sediment deposition. The text now reads (L198-200): "Plotting the residuals to the best-fit plane along UTM northing reveals some structure in the most downstream southern long profile extent (Fig. 4A insert), likely due to Holocene sedimentation (Blum et al., 1995; Blum and Aslan, 2006)."

Line 192-194: I suggest to move this sentence up to line 189 to state from the beginning, why this analysis is done.

We have moved the suggested text up in the paragraph (L195-196). The text now reads "The best-fit plane for the modern valley was used to generate detrended elevations for each terrace DEM measurement by subtracting it from the spatially corresponding modern valley best-fit plane value."

Figure 3: It would be helpful to color the datapoints in (A) and (B) according to the terrace they belong to.

We have added the grouping to points in Figure 3.

Figure 6: The actual RMSE values for the three terraces are not give here, they only come up later in section 4. Please briefly give the values already when describing the fits in the results.

We have updated the figure caption to read (L241): "The low, intermediate, and high Deweyville terrace sets have RMSEs of 1.43m, 1.54m, and 1.41m, respectively."

Line 506: Remove 'introduce'?

We thank the reviewer for catching this mistake. The text now reads (L514-516): "We suggest that paleo-channel characteristics are a more faithful record of discharge changes in fluvial systems and that additional bend metrics can differentiate autogenic terrace formation processes, specifically bend cut-off from unsteady lateral migration rates."